# A New Test Method for Evaluation of Solidification Cracking Susceptibility of Stainless Steel during Laser Welding

**DOI:** 10.3390/ma13143178

**Published:** 2020-07-16

**Authors:** Wenbin Wang, Li Xiong, Dan Wang, Qin Ma, Yan Hu, Guanzhi Hu, Yucheng Lei

**Affiliations:** 1School of Materials Science and Engineering, Jiangsu University, Zhenjiang 212013, China; 2221805080@stmail.ujs.edu.cn (W.W.); alley_xl@163.com (L.X.); Xiaomaer_70@126.com (Q.M.); sys936452544@163.com (Y.H.); hugzing@163.com (G.H.); yclei@ujs.edu.cn (Y.L.); 2Key Laboratory of High-end Structural Materials of Jiangsu Province, Jiangsu University, Zhenjiang 212013, China; 3Key Laboratory of Agricultural Machinery Equipment Remanufacturing Technology of Jiangsu Province, Jiangsu University, Zhenjiang 212013, China

**Keywords:** trapezoidal hot cracking test, solidification cracking susceptibility, stainless steel, laser welding

## Abstract

A new test method named “Trapezoidal hot” cracking test was developed to evaluate solidification cracking susceptibility of stainless steel during laser welding. The new test method was used to obtain the solidification cracking directly, and the solidification cracking susceptibility could be evaluated by the solidification cracking rate, defined as the ratio of the solidification cracking length to the weld bead length under certain conditions. The results show that with the increase in the solidification cracking rate, the solidification cracking susceptibility of SUS310 stainless steel was much higher than that of SUS316 and SUS304 stainless steels during laser welding (at a welding speed of 1.0 m/min) because a fully austenite structure appeared in the weld joint of the former steel, while the others were ferrite and austenitic mixed structures during solidification. Besides, with an increase in welding speed from 1.0 to 2.0 m/min during laser welding, the solidification cracking susceptibility of SUS310 stainless steel decreased slightly; however, there was a tendency towards an increase in the solidification cracking susceptibility of SUS304 stainless steel due to the decrease in the amount of ferrite under a higher cooling rate.

## 1. Introduction

Laser welding has already been employed in the various processing and manufacturing fields due to the advantages of fast welding speed, high energy density, small welding deformation and so on [1,2,3]. Solidification cracking susceptibility could be changed during laser welding with a relatively higher welding speed compared to that of gas tungsten arc welding (GTAW), since the susceptibility can be affected by solidification behavior which depends on not only chemical composition but also on welding conditions, such as the welding speed considered as a very important factor. However, research on the solidification cracking susceptibility during laser welding is rare.

At present, the test methods for evaluating the solidification cracking susceptibility include mainly two categories. One is an external restraint test method, like Varestraint test [4,5,6], Sigmajig test [7,8], Rapid tensile test [9,10], etc. The other category is self-restraint test methods, like Houldcroft cracking test [11,12], FISCO hot cracking test [13,14], Circular patch test [15,16], etc. Traditionally, the GTAW is used as a heat source in the above test methods to evaluate the solidification cracking susceptibility. While there are few test methods to evaluate the susceptibility during laser welding, Shinozaki [17,18,19,20,21,22] et al. developed a U type hot cracking test based on laser welding and laser Trans-Varestraint test [17,18,19,20,21,22] to evaluate solidification cracking susceptibility of stainless steel quantitatively. Nevertheless, these methods are difficult to adopt for widespread use, since it is hard to measure temperature accurately during laser welding at high welding speed and also it needs expensive equipment, such as the Varestraint test machine, high-speed camera, and high precision temperature measurement device and so on. The solidification cracking susceptibility of aluminum alloy was evaluated quantitatively using the Houldcroft test during laser welding with filler wire by Zhai [23]. However, it cannot be applied to evaluate the solidification cracking susceptibility of stainless steel using the Houldcroft test during laser welding. Because the size of weld bead and thermal conductivity are both relatively smaller compared with that of aluminum alloy welding, the solidification cracking could not initiate for stainless steel during laser welding using the Houldcroft test. Fukuhisa [24] et al. developed a novel test method based on electron beam welding to evaluate the solidification cracking susceptibility of not only aluminum alloy but also stainless steel. However, the research on the solidification cracking susceptibility of stainless steel during laser welding is still rather limited. Thus, it is necessary to develop a relatively simple and widely used test method to evaluate the solidification cracking susceptibility during laser welding. 

Certainly, the advantages of the new test method should be obvious compared with the previous methods. Firstly, it should have a low requirement on the equipment, that is, it is better to only need a heat source and some simple specimen to evaluate the solidification cracking susceptibility. Besides, the heat source of this new test method could be changed easily and made applicable not only for a laser, but also for the others, like GTAW. Thirdly, the new test method may evaluate the solidification cracking susceptibility of different kinds of materials, such as stainless steel or light alloy, etc. In brief, the developed new test method should have the characteristics of the simple equipment, wide application, low cost and reliability. Therefore, the purpose of this research is to develop a new test method owning the advantages mentioned above to evaluate the effect of the chemical composition on the solidification cracking susceptibility of stainless steel during laser welding at different welding speeds.

## 2. Materials and Experiment 

### 2.1. Materials

SUS310, SUS316 and SUS304 austenitic stainless steels with a thickness of 2 mm, were used for experimentations. Initially, SUS310 stainless steel was employed to develop the novel test method for the evaluation of solidification cracking susceptibility. The chemical composition of these stainless steels is shown in Table 1.

### 2.2. Experiment

Table 2 shows the laser welding conditions. Figure 1 shows the experimental equipment and the respective method. A fiber laser with a 1064-nm wavelength was employed as a heat source. To obtain the full penetration, the welding speeds equivalent to 1.0, 1.5 and 2.0 m/min and the corresponding laser powers equal to 1.8 kW, 2.3 kW and 2.8 kW, respectively, were utilized. Moreover, the new test method was developed when the laser welding speed was 1.0 m/min. The laser spot size was 0.6 mm (just focus) and the laser head was tilted 10 degrees towards the welding direction in order to prevent the laser beam reflecting into the fiber and damaging the instrument during laser welding. The front and backside of specimens were shielded during laser welding by Argon with flow rates of 20 L/min, respectively. In addition, the final edge of the specimen was installed with fixtures so that the welding direction was always along the centerline of the specimen during laser welding, as shown in Figure 1. 

After experimentation, the specimens were manufactured using a wire-cutting machine, and set with the cold inserts to stand conveniently for observation of the microscopic structures. The microstructure was characterized through an optical microscope (OM), a scanning electron microscope (SEM), and an electron back scatter diffraction (EBSD). Some specimens for the observations of OM and SEM were corroded with ferric chloride aqueous solution, the ratio of which is HCl solution: water: FeCl_3_ solid = 30 mL: 120 mL: 10 g. The corrosion time of each weld specimen is 5–10 seconds and the others were prepared using the electrochemical polishing for the observation of EBSD.

## 3. Results and Discussion

### 3.1. Principle of the New Test Method—“Trapezoidal hot” Cracking Test

Figure 2 shows the schematic image of the specimen for the new test method to evaluate the solidification cracking susceptibility during laser welding. The specimen geometry for this new test method is the same as a trapezoid, so it is called the “Trapezoidal hot” cracking test, as shown in Figure 2a. The key behind this trapezoidal test is to decide the values of W_S_ and W_F_ which are defined as the starting and final edges of the welding process, respectively. To ensure that the full penetration weld bead can be obtained, during laser welding (especially at the start), the wedge groove of 1 mm × 5 mm was machined out at the starting edge of the specimen, as shown in Figure 2b. 

The “Trapezoidal hot” cracking test belongs to the self-restraint test method category, which normally includes three steps, initiation, propagation, and ceasing of the solidification cracking, respectively, to evaluate the solidification cracking susceptibility. Thus, it is essential to clarify the influence of each step in the case of the “Trapezoidal hot” cracking test. Importantly, the “Trapezoidal hot” cracking test would not just need to initiate the solidification cracking during welding, but would also need to cease the cracking before the end of the welding, because it is impossible to evaluate and compare the susceptibility if there is no crack or the crack appears throughout the weld bead during each test. For this reason, the values of W_S_ and W_F_ play an important role in initiating, propagating, and ceasing the solidification cracking, respectively.

During welding, the relatively higher weld stress will occur at the rear of the melt pool perpendicular to the direction of the temperature gradient as a result of thermal expansion produced along with the centerline of the weld bead. In the “Trapezoidal hot” cracking test, solidification cracking could be initiated by a high rate of tearing strain produced by a rapid thermal expansion of the weld bead. If W_S_ is too narrow, the solidification cracking will not initiate because of the relatively lower deflection force as a result of the lack of temperature gradient. If W_S_ is too wide, the cracking will also not initiate because of the relatively higher restraint as a result of the wide specimen width. Therefore, determining the value of W_S_ is an important step in the “Trapezoidal hot” cracking test [24].

Subsequently, the solidification cracking will propagate due to the phenomenon that the difference between the deflection force and the restraint exceeds that of critical stress in the brittle temperature range. With the increase in the specimen width along the welding direction, the restraint will gradually increase. When the restraint is higher than the critical value, the solidification cracking will increase. Thus, it is also an essential step to optimize the value of W_F_ in the “Trapezoidal hot” cracking test. Finally, the solidification cracking susceptibility could be evaluated under the appropriate values of W_S_ and W_F_ using the “Trapezoidal hot” cracking test during laser welding.

### 3.2. Development of the “Trapezoidal hot” Cracking Test

Figure 3 shows the method to obtain a suitable value of W_S_. The size of the specimen is 100 mm *×* 100 mm *×* 2 mm. Firstly, the distance from the specimen edge is defined as (1/2)W_S_, which is the starting position of the welding, as shown in Figure 3a. Figure 4 shows the macro profile of the solidification cracking under the different values of (1/2)W_S_ during laser welding at a welding speed of 1.0 m/min. Figure 5 indicates the solidification cracking length under the different values of (1/2)W_S_ during laser welding at a welding speed of 1.0 m/min corresponding to those of Figure 4. When the value of (1/2)W_S_ was less than 6 mm, the solidification cracking appeared, and the length was more than 80 mm at both the front and backside of the specimen. Furthermore, there was no crack under the value of (1/2)W_S_ more than 7 mm, as shown in Figure 4 and Figure 5. According to the above principle, the value of W_S_ could be set 12 mm to initiate cracking at the start of the welding.

Then, the value of W_F_ was assumed as 20 mm, 30 mm, 40 mm, and 50 mm, respectively. Figure 6 shows the macroscopic appearance of the solidification cracking of SUS310 stainless steel under W_S_ of 12 mm and the different values of W_F_ during laser welding at a welding speed of 1.0 m/min. The red arrow indicates the ceasing point of the cracking, as shown in Figure 6. Figure 7 shows the effect of W_F_ on the solidification cracking length corresponding to those in Figure 6. The solidification cracking length tended to decrease with increasing the value of W_F_.

Actually, the solidification cracking susceptibility could not be changed by the value of W_F_. However, under a certain condition, the solidification cracking susceptibility can be characterized by the cracking length. Reddy [25] et al. indicated that the susceptibility of the solidification cracking could be evaluated by using the cracking length of the specimen after welding. The longer the cracking length was, the higher the cracking susceptibility tended to be, respectively. Here, the solidification cracking rate calculated by the ratio of the solidification cracking length to the weld bead length would be defined as an important index to evaluate the solidification cracking susceptibility under the suitable values of W_S_ and W_F_.

### 3.3. The Effect of the Chemical Composition on the Solidification Cracking Susceptibility of Stainless Steel During Laser Welding at a Welding Speed of 1.0 m/min

It is necessary to evaluate the effect of (*1/2*)W_S_ on the solidification cracking length of SUS304 and SUS316 stainless steels during laser welding at a welding speed of 1.0 m/min in order to define the true W_S_ because of the different chemical components. When the values of (*1/2*)W_S_ were 6 mm and 7 mm for SUS304 and SUS316 stainless steels, respectively, a relatively small crack initiated, as shown in Figure 8. Therefore, in order to ensure the initiation of the solidification cracking during laser welding for different stainless steels, the value of (*1/2*)W_S_ should be defined as 6 mm. Figure 9 shows the effect of the chemical components (different stainless steels) on the solidification cracking rate during laser welding at a welding speed of 1.0 m/min. The sequence of the solidification cracking rate under W_S_ of 12 mm and W_F_ of 40 mm is as follows: SUS310 > SUS316 > SUS304. Especially, the solidification cracking rate of SUS310 stainless steel was much higher than the others. 

The microstructure of the cross-section of the weld joint for each material is shown in Figure 10. Plenty of dendrites were distributed in the cross-section and grew from the sides of the weld bead to the centerline along the easiest direction. The structure was fully austenite during the solidification of SUS310 stainless steel welding, as shown in Figure 10a,d. In the austenite, some lath ferrite structures could be observed from the microstructure images of SUS316 and SUS304 stainless steels welding; further, the amount of ferrite structure in SUS304 stainless steel seemed to be more than that of SUS316 stainless steel, as shown in Figure 10b,c,e,f.

According to the welding metallurgy of stainless steel and previous researches, the microstructures of the weld bead are dependent on the solidification mode [26]. At the same time, the value of Cr_eq_/Ni_eq_ can determine the solidification mode of the welding. Therefore, Cr_eq_/Ni_eq_ is an important factor in the evaluation of the solidification cracking susceptibility, and it can be calculated as follows.
Cr_eq_ = Cr +Mo + 1.5Si + 0.5Nb(1)
Ni_eq_ = Ni + 30C + 0.5Mn(2)

If the value of Cr_eq_/Ni_eq_ is lower than 1.25, the stainless steel solidifies in fully austenitic mode (A-mode), that is, the microstructure is fully austenitic at the end of solidification. If the value of Cr_eq_/Ni_eq_ is between 1.25 and 1.48, the initial phase is austenite, and then δ-ferrite is formed by reaction, so it is called AF-mode. Since the ferrite forming elements, such as Cr, can segregate in the subgrain boundary to promote the formation of ferrite, the cracking susceptibility could be reduced to a certain extent. If the value of Cr_eq_/Ni_eq_ is between 1.48 and 1.95, FA-mode occurs during the solidification. The ferrite is firstly precipitated from the liquid phase and then the austenite forms and experiences a peritectic–eutectic reaction at the end of solidification. Because the primary ferrite can solute higher impurities, like S and P, this mode is beneficial to restrict the partitioning of these elements to the interdendritic regions, causing a good resistance to the solidification cracking. If the value of Cr_eq_/Ni_eq_ is greater than 1.95, it forms the full ferrite during the solidification [27]. 

Through calculation, the values of Cr_eq_/Ni_eq_ in SUS310, SUS316, and SUS304 stainless steels were 1.11, 1.53, and 1.91, respectively. Therefore, SUS310 stainless steel solidified in A-mode, while SUS316 and SUS304 stainless steels solidified in FA-mode. The fully austenitic structure contributed to the greater susceptibility compared with ferrite and austenitic mixed structure, as shown in Figure 10. SUS316 and SUS304 stainless steels usually solidified in ferrite + austenite mode, and the metallographic structure is usually lath ferrite + austenite. Figure 11 shows the EBSD measurement results of the microstructure of the cross-section of weld joint of SUS304 stainless steel at the welding speed of 1.0 m/min. During solidification, the grain grows along the easy growth direction perpendicular to the solid–liquid interface and parallel to the heat flow direction. In the weld metal of SUS304 stainless steel, the microstructure included plenty of austenite and a small amount of ferrite distributed along the grain boundary of the austenite, which agrees with the results of OM and SEM.

When solidifying in FA-mode, this excellent crack resistance is due to the mixed structure of austenite + ferrite along the solidified grain boundary at the end of the solidification, which further prevents the infiltration of the liquid film and enables the crack along the generated curved grain boundary extension [28], ultimately resulting in the lower cracking susceptibility of SUS316 and SUS304 stainless steels. Moreover, the solidification cracking susceptibility of SUS304 stainless steel was low compared with that of SUS316 stainless steel because of the appearance of relatively more lath ferrite.

### 3.4. The Effect of Laser Welding Speed on the Solidification Cracking Susceptibility of Stainless Steel

Figure 12 shows the effect of welding speed on the solidification cracking rate of SUS310 and SUS304 stainless steels during laser welding using the “Trapezoidal hot” cracking test with W_S_ of 12 mm. For SUS310 stainless steel, the solidification cracking rate was assessed under W_F_ of 40 mm during laser welding at different welding speeds ranging from 1.0 to 2.0 m/min, since the cracking rate was already high during laser welding at a welding speed of 1.0 m/min under W_F_ of 20 mm. Besides, due to the relatively low solidification cracking rate in SUS304 stainless steel compared with that of SUS310 stainless steel, the value of W_F_ was set to 20 mm to measure and compare the cracking rate conveniently. The results showed that with the increase in laser welding speed, the solidification cracking rate of SUS310 stainless steel tended to decrease gradually, that is, the cracking susceptibility dropped slowly in the case of A-mode solidification. In contrast, there was a tendency for the solidification cracking rate of SUS304 stainless steel to increase according to the increase in welding speed, that is, the cracking susceptibility augmented in the case of FA-mode solidification.

The microstructures of the cross-section of SUS310 and SUS304 stainless steels during laser welding at the welding speeds from 1.0 to 2.0 m/min are shown in Figure 13. The grains in SUS310 stainless steel had a certain degree of refinement during the increase in welding speed, as shown in Figure 13a,b. While, for SUS304 stainless steel during laser welding, the amount of δ-ferrite in the microstructure was less at a welding speed of 2.0 m/min than those at 1.0 m/min, as shown in Figure 13c,d. 

Theoretically, when the power is fixed during laser welding, the welding heat input, calculated using the power divided by the welding speed, will decrease with the increase in welding speed, resulting in the appearance of a relatively higher cooling rate, and further causing grain refinement [29]. Furthermore, according to the calculations the heat inputs were 108 J/mm for a welding speed of 1.0 m/min, 92 J/mm for a welding speed of 1.5 m/min and 84 J/mm for a welding speed of 2.0 m/min. Thus, the cooling rate tended to increase with an increase in welding speed from 1.0 to 2.0 m/min in this experiment during laser welding, inducing the phenomenon that the grain size was reduced. During solidification, the grain refinement can not only promote the refill and healing of liquid phase more effectively but also increase the number of grain boundaries, further preventing harmful low melting-point compounds from segregating on grain boundaries [30]. Moreover, Agarwal [31] et al. pointed out that during SUS310 stainless steel laser welding, with the increase in welding speed, the solidification segregation was inhibited and the brittleness temperature range was reduced, resulting in a decrease in the solidification cracking susceptibility. Therefore, there was a decrease slightly for SUS310 stainless steel during laser welding with an increase in welding speed from 1.0 to 2.0 m/min. Inversely, because of a good performance in inhibiting segregation for δ-ferrite, the solidification cracking susceptibility would increase with the decrease in the amount of δ-ferrite for SUS304 stainless steel during laser welding as the welding speed increases from 1.0 m/min to 2.0 m/min.

## 4. Conclusions

This research mainly developed the “Trapezoidal hot” cracking test to evaluate the solidification cracking susceptibility of stainless steel during laser welding at different welding speeds. The conclusions are as follows:(1)The new test method called the “Trapezoidal hot”cracking test was developed to evaluate the solidification cracking susceptibility using the solidification cracking rate which was the ratio of the solidification cracking length to the weld bead length under the suitable values of W_S_ and W_F_.(2)The sequence of the solidification cracking susceptibility was as follows: SUS310 > SUS316 > SUS304 during laser welding at a welding speed of 1.0 m/min; because the fully austenitic structure appeared in the weld joint of SUS310 stainless steel, and the ferrite and austenitic mixed structures were produced in SUS316 and SUS304 stainless steel, the latter had relatively more ferrite.(3)With the increase in welding speed from 1.0 to 2.0 m/min, the solidification cracking susceptibilities of SUS310 and SUS304 stainless steels tended to decrease and increase during laser welding, respectively. This is due to the grain refinement for the former and a decrease in the amount of ferrite for the latter.

## Figures and Tables

**Figure 1 materials-13-03178-f001:**
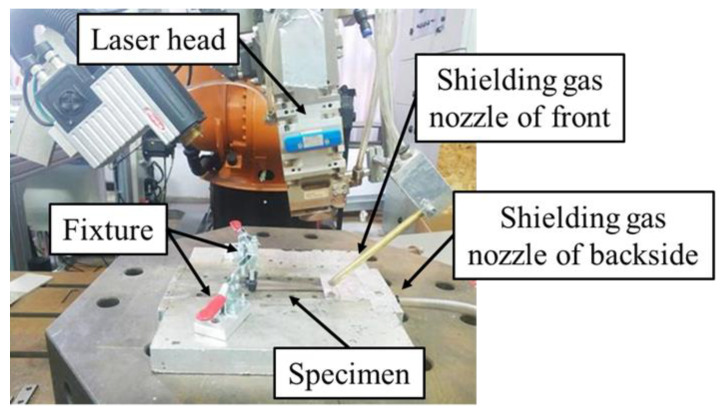
Experimental equipment and method.

**Figure 2 materials-13-03178-f002:**
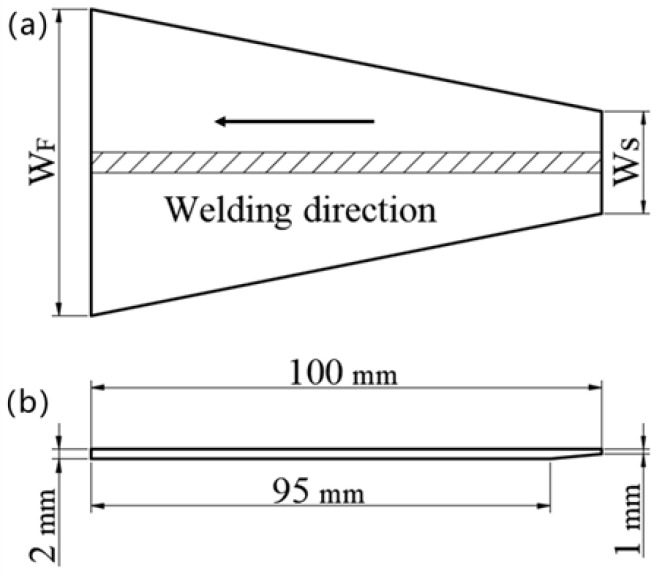
Schematic image of the specimen of the new test method—“Trapezoidal hot” cracking test. (**a**) The front of the specimen; (**b**) the profile of the specimen.

**Figure 3 materials-13-03178-f003:**
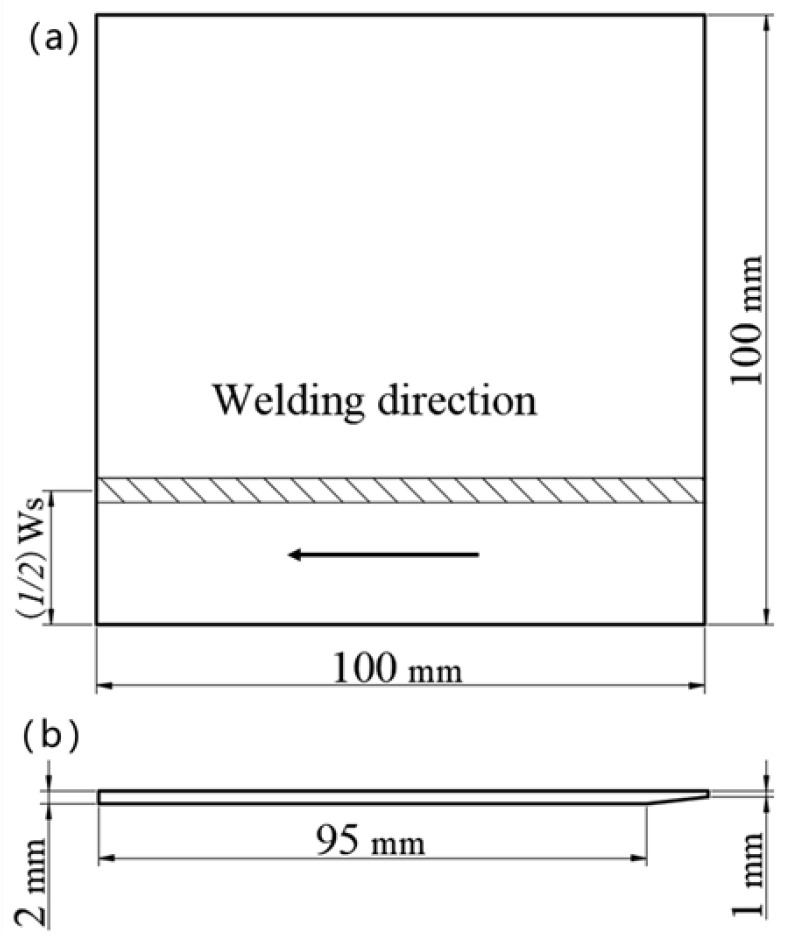
Method to determine the value of the starting edge of the welding process (W_S_). (**a**) The front of the specimen; (**b**) the profile of the specimen.

**Figure 4 materials-13-03178-f004:**
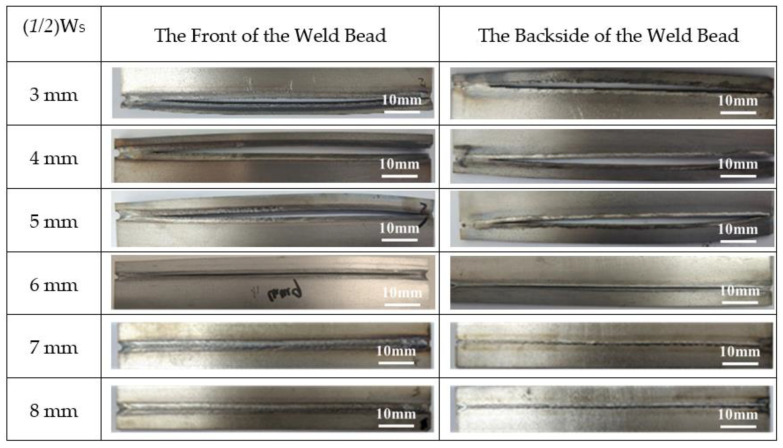
The macro profile of the solidification cracking of SUS310 stainless steel under the different values of (*1/2*)W_S_ during laser welding at a welding speed of 1.0 m/min.

**Figure 5 materials-13-03178-f005:**
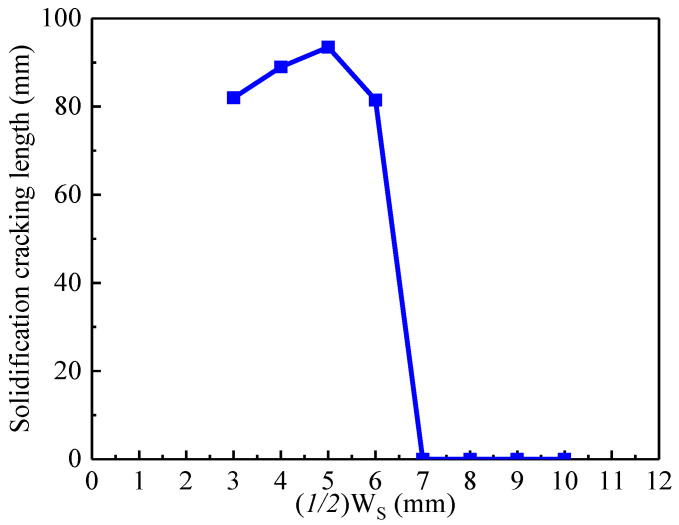
Effect of (*1/2*)W_S_ on the solidification cracking length during laser welding at a welding speed of 1.0 m/min.

**Figure 6 materials-13-03178-f006:**
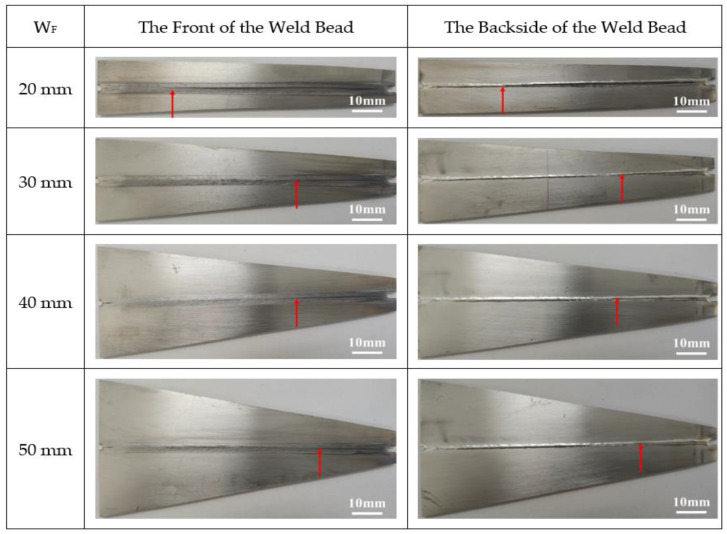
Macroscopic appearance of the solidification cracking of SUS310 stainless steel under W_S_ of 12 mm and the different values for the final edge of the welding process (W_F_) during laser welding at a welding speed of 1.0 m/min.

**Figure 7 materials-13-03178-f007:**
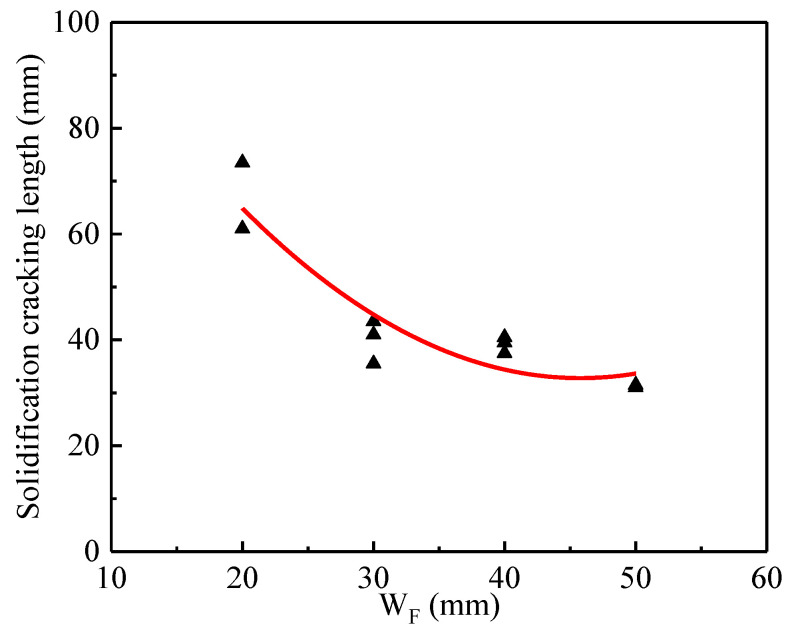
Effect of W_F_ on the solidification cracking length under W_S_ of 12 mm during laser welding at a welding speed of 1.0 m/min.

**Figure 8 materials-13-03178-f008:**
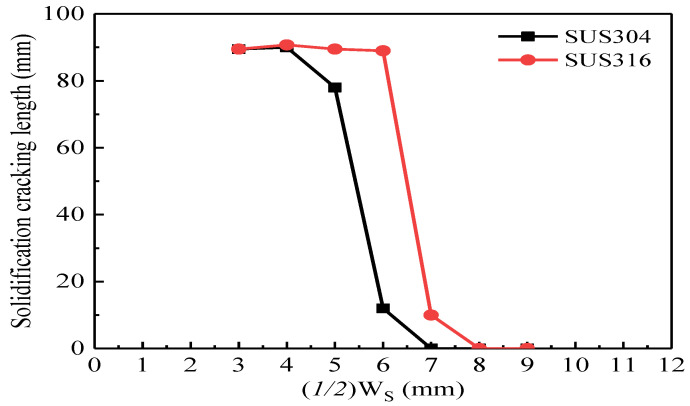
Effect of (*1/2*)W_S_ on the solidification cracking length of the SUS316 and SUS304 stainless steels during laser welding at a welding speed of 1.0 m/min.

**Figure 9 materials-13-03178-f009:**
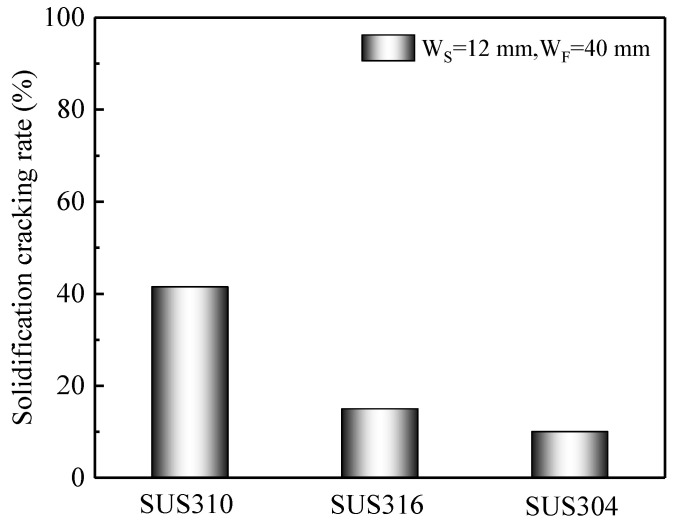
Effect of the chemical components on the solidification cracking rate during laser welding at the welding speed of 1.0 m/min.

**Figure 10 materials-13-03178-f010:**
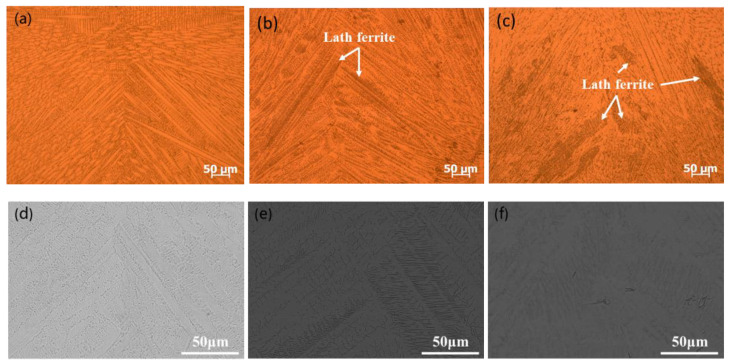
Optical microscope (OM) and scanning electron microscope (SEM) images of the microstructure of the cross-section of the weld joint during laser welding at a welding speed of 1.0 m/min. (**a**) OM image in SUS310; (**b**) OM image in SUS316; (**c**) OM image in SUS304; (**d**) SEM image in SUS310; (**e**) SEM image in SUS316; (**f**) SEM image in SUS304.

**Figure 11 materials-13-03178-f011:**
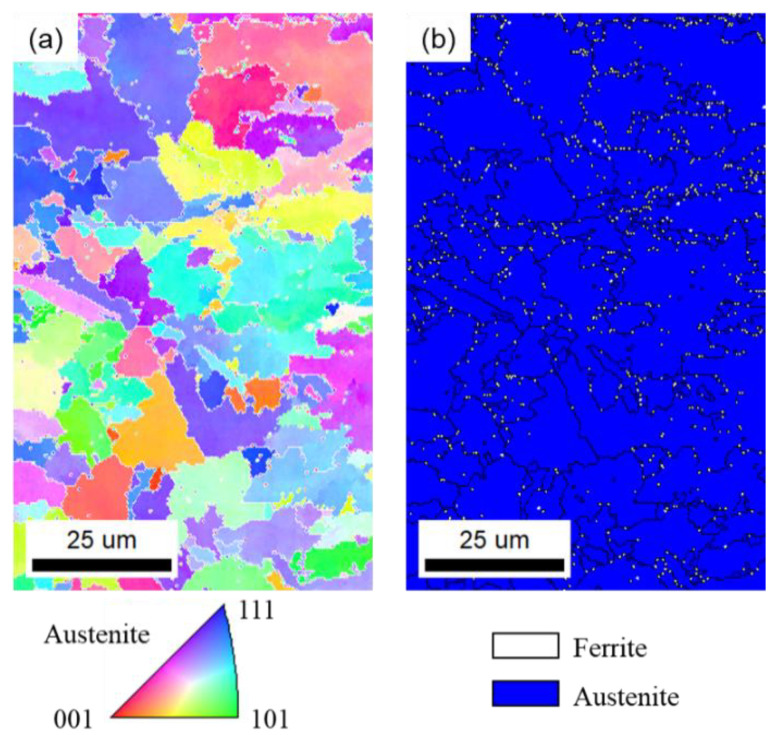
Electron back scatter diffraction (EBSD) measurement results of the microstructure of the cross-section of the weld joint of SUS304 stainless steel during laser welding at the welding speed of 1.0 m/min. (**a**) Orientation color map with the color legend; (**b**) phase mapping with the grain boundary.

**Figure 12 materials-13-03178-f012:**
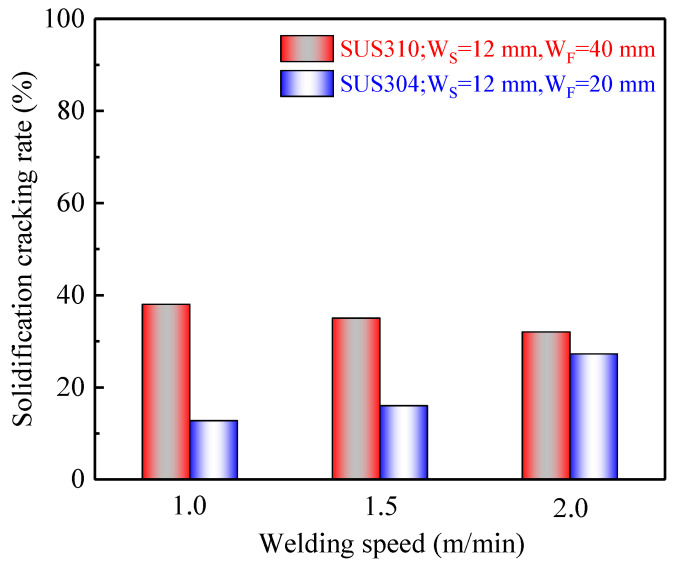
Effect of the welding speeds on the solidification cracking rate during laser welding.

**Figure 13 materials-13-03178-f013:**
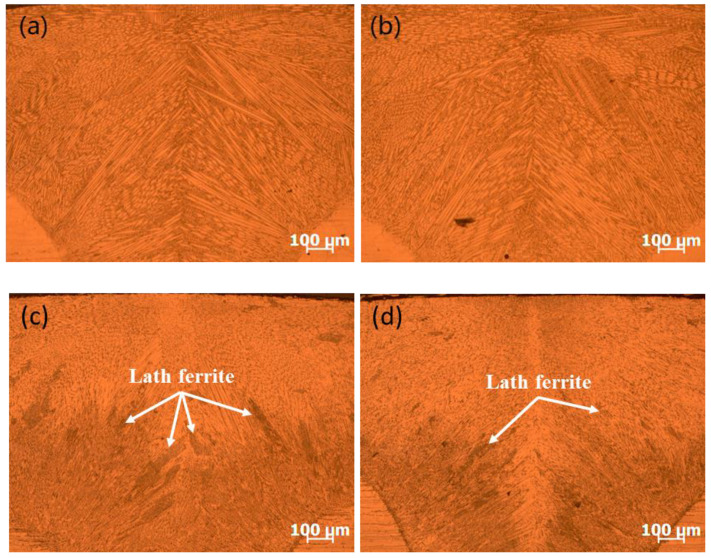
OM image of the microstructure of the cross-section of the weld joint. (**a**) OM image in SUS310 at the welding speed of 1.0 m/min; (**b**) OM image in SUS310 at the welding speed of 2.0 m/min; (**c**) OM image in SUS304 at the welding speed of 1.0 m/min; (**d**) OM image in SUS304 at the welding speed of 2.0 m/min.

**Table 1 materials-13-03178-t001:** Chemical composition of stainless steels (wt. %).

Materials	C	Si	Mn	P	S	Cr	Ni	Mo	Cu	Fe
SUS310	0.07	0.40	0.95	0.015	0.014	24.7	20.3	0.12	0.10	Bal.
SUS316	0.06	0.26	1.12	0.015	0.019	16.7	10.2	2.11	0.09	Bal.
SUS304	0.04	0.32	1.04	0.015	0.016	18.3	8.1	0.01	0.02	Bal.

**Table 2 materials-13-03178-t002:** Laser welding conditions.

Welding Speed, m/min	1.0	1.5	2.0
Laser power, kW	1.8	2.3	2.8
Laser spot size, mm	0.6 (Just focus)		
Laser irradiation angle, deg	10		
Ar shielding gas, L/min	20 (Front and backside)		

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
