# Peer review of "A New Test Method for Evaluation of Solidification Cracking Susceptibility of Stainless Steel during Laser Welding"

_materials, 2020, doi:10.3390/ma13143178_

Round 1
Reviewer 1 Report
In this work, authors introduced an experimental technique to evaluate the solidification cracking susceptibility by using the solidification rate. In the opinion of the reviewer, the approach is indeed interesting. However, authors must provide a solid and concrete reason about why the community need this new experimental technique. What will be the main advantages over other testing methods? Below, author could find more comment in the manuscript:
- In the introduction, authors provided a list of equipment for evaluating the solidification cracking susceptibility and disadvantages for each category. It would also better to point out how could the proposed technique be better than existing ones in the introduction to motivate reader to follow this technique of authors.
- In line 99, use “which” instead of “who”
- In line 102, “was machined” not “was be machined”
- In line 107-108, “to clarify the effect of each step of the Trapezoidal hot cracking test,”
- In line 109, “would not”
- In line 131, “x2mm. Firstly, the distance …”
- Authors determined W_s and W_f from the experimental results of SUS310. In the opinion of reviewer, authors should also perform similar experimental set up on other materials in order to define the true W_s and W_f.
- In line 221-222, “since the cracking rate was already high during…”
- In Figure 11, please highlight the delta ferrite in the microstructure
- In terms of microstructure characterization, authors mainly used LOM and a simple SEM for analyzing microstructure morphology. For a research in 2020, there are many more modern microstructure characterization tool which allow an in-depth analysis of microstructural features like EBSD. Would it be possible for authors to perform an EBSD on the weldment?
- For the LOM analysis, please also mention about the etching solution used and how specimens were prepared.
Author Response
Response to Reviewer 1 Comments
Thank you very much for your comments. The authors also upload an attachment including every response and some added figures.
Point 1: In this work, authors introduced an experimental technique to evaluate the solidification cracking susceptibility by using the solidification rate. In the opinion of the reviewer, the approach is indeed interesting. However, authors must provide a solid and concrete reason about why the community need this new experimental technique.
Response 1: Thank you for your comments. Those comments are all valuable and very helpful for revising and improving our manuscript, as well as the important guiding significance to our researches. We have studied comments carefully and have made the corresponding correction.
Although the testing methods to evaluate solidification cracking susceptibility have a lot, such as Trans-varestraint test, Sigmajig test, Rapid tensile test, etc., the corresponding heat source is TIG. For laser, due to high welding speed and small weld bead, it is different to test some data for evaluation. In addition, it needs the complex and expensive experiment equipment, such as a high-speed camera. More importantly, it is very hard to measure the precise temperature profile to evaluate the cracking susceptibility using a thermocouple or an optical fibre radiation thermometer, etc. Also, it requires more than one people, like two or three people, to do these tests together. Moreover, the experimental period is relatively long compared with our new test method. It is known that the final target of the research is to apply to production. For the enterprise, it is very important to save the time costs, money costs and human costs. As for the experimental method mentioned in our paper, it adopts a relatively simple and common experimental equipment. It only needs one people to do the test for evaluation. Meanwhile, the heat source of this experimental method is applicable not only for a laser, but also for the other, like TIG. Thus, we think it is necessary to study our new experimental technique.
Point 2: What will be the main advantages over other testing methods?
Response 2: Special thanks to you for your good comments.
Authors think that the main advantages of the “Trapezoidal hot” cracking test are as follows: first of all, it only needs simple experimental equipment, like a heat source and some trapezoidal specimens. Secondly, this method is not limited to laser welding, but also to the other welding heat sources. Thirdly, the test method can be applied to other materials for evaluating the solidification cracking susceptibility theoretically. Importantly, this method is appropriate for the high welding speed.
Point 3: In the introduction, authors provided a list of equipment for evaluating the solidification cracking susceptibility and disadvantages for each category. It would also better to point out how could the proposed technique be better than existing ones in the introduction to motivate reader to follow this technique of authors.
Response 3: Thank you again for your valuable suggestions.
The detailed revision is as follow:
In line 62, “The purpose of this research is to develop a new test method to evaluate the effect of the chemical composition on the solidification cracking susceptibility of stainless steel during laser welding at different welding speeds.” has been changed into “Certainly, the advantages of the new test method would be obvious compared with the previous methods. Firstly, it should have a low requirement on the equipment, that is, it is better to only need a heat source and some simple specimen to evaluate the solidification cracking susceptibility. Besides, the heat source of this new test method could be changed easily and applicable not only for a laser, but also for the others, like GTAW. Thirdly, the new test method may evaluate the solidification cracking susceptibility of kinds of materials, such as stainless steel or light alloy, etc. In a word, the developed new test method should have the characteristic of the simple equipment, widely application, low cost and reliable. Therefore, the purpose of this research is to develop a new test method owning the advantages mentioned above to evaluate the effect of the chemical composition on the solidification cracking susceptibility of stainless steel during laser welding at different welding speeds.”
Point 4: In line 99, use “which” instead of “who”
Response 4: The detailed revision is as follow:
In line 99, “… to decide the values of WS and WF who are …” has been changed into “… to decide the values of WS and WF which are …”
Point 5: In line 102, “was machined” not “was be machined”
Response 5: The detailed revision is as follow:
In line 102, “… was be machined out ...” has been changed into “… was machined out ...”
Point 6: In line 107-108, “to clarify the effect of each step of the Trapezoidal hot cracking test,”
Response 6: The detailed revision is as follow:
In line 107-108, “Thus, it is necessary to make clear the effect of the each step in the case of the trapezoidal hot cracking test.” has been changed into “Thus, it is necessary to clarify the effect of each step of the “Trapezoidal hot” cracking test.”
Point 7: In line 109, “would not”
Response 7: The detailed revision is as follow:
In line 109, “… wouldn’t just need to initiate the solidification cracking during welding …” has been changed into “… would not just need to initiate the solidification cracking during welding ...”
Point 8: In line 131, “x2mm. Firstly, the distance …”
Response 8: The detailed revision is as follow:
In line 131, “… × 2 mm, firstly. The distance from the specimen edge is …” has been changed into “… × 2 mm. Firstly, the distance from the specimen edge is ...”
Point 9: Authors determined WS and WF from the experimental results of SUS310. In the opinion of reviewer, authors should also perform similar experimental set up on other materials in order to define the true WS and WF.
Response 9: Thank you very much.
The authors had already test the WS of SUS304 and SUS316 stainless steels using the similar experiment. We have put the previous data on the manuscript. When the values of (1/2)WS are 6 mm and 7 mm for SUS304 and SUS316 stainless steels, respectively, the relatively small crack initiated. While, in order to ensure the initiation of the solidification cracking during laser welding for different stainless steels, the value of (1/2)WS should be defined as 6 mm. Thus, the true WS for these three stainless steels is 12 mm. For the WF, the authors think that the Figures 7, 8 and 10 in the original manuscript have shown influence. In authors opinion, for the materials with high cracking susceptibility, the value of WF might be relatively high in order to prevent the crack throughout the weld bead; while, for the one with low cracking susceptibility, the value of WF might be relatively small in order to reduce the measurement error because of the appearance of the shorter solidification cracking length.
The detailed revision is as follow:
In line 169, “Figure 8 shows the effect of the chemical components ...” has been changed into “It is necessary to evaluate the effect of (1/2)WS on the solidification cracking length of SUS304 and SUS316 stainless steels during laser welding at a welding speed of 1.0 m/min in order to define the true WS because of the different chemical components. When the values of (1/2)WS were 6 mm and 7 mm for SUS304 and SUS316 stainless steels, respectively, the relatively small crack initiated, as shown in Figure 8. Therefore, in order to ensure the initiation of the solidification cracking during laser welding for different stainless steels, the value of (1/2)WS should be defined as 6 mm. Figure 9 shows the effect of the chemical components ...”
Point 10: In line 221-222, “since the cracking rate was already high during…”
Response 10: The detailed revision is as follow:
In line 221-222, “… since the cracking rate was already the relatively high during laser welding at welding speed of 1.0 m/min under WF of 20 mm.” has been changed into “… since the cracking rate was already high during laser welding at a welding speed of 1.0 m/min under WF of 20 mm.”
Point 11: In Figure 11, please highlight the delta ferrite in the microstructure
Response 11: Thanks a lot for your valuable comments. And the delta ferrite has been highlighted in the corresponding figure.
Point 12: In terms of microstructure characterization, authors mainly used LOM and a simple SEM for analysing microstructure morphology. For a research in 2020, there are many more modern microstructure characterization tool which allow an in-depth analysis of microstructural features like EBSD. Would it be possible for authors to perform an EBSD on the weldment?
Response 12: It is very kind of you to give us a meaningful suggestion using modern microstructure characterization tool to analyze the microstructural features. We have adopt this comment on our manuscript. The distribution of austenite and ferrite in SUS304 stainless steel was characterized by EBSD. The results show that the microstructure of the cross-section of weld joint SUS304 stainless steel included plenty of austenite and a small amount of ferrite distributed along the grain boundary of the austenite.
The detailed revision is as follow:
In line 205, “When solidifying in FA-mode …” has been changed into “Figure 11 shows the EBSD measurement results of the microstructure of the cross-section of weld joint of SUS304 stainless steel at the welding speed of 1.0 m/min. During solidification, the grain grows along the easy growth direction perpendicular to the solid-liquid interface and parallel to the heat flow direction. In the weld metal of SUS304 stainless steel, the microstructure included plenty of austenite and a small amount of ferrite distributed along the grain boundary of the austenite, which agrees with the results of OM and SEM. When solidifying in FA-mode …”
Point 13: For the LOM analysis, please also mention about the etching solution used and how specimens were prepared.
Response 13: The detailed revision is as follow:
In line 83-85, “After test, the specimens were manufactured using a wire-cutting machine and the metallographic specimens were corroded with ferric chloride aqueous solution. The microstructure was characterized through an optical microscope (OM) and a scanning electron microscope (SEM).” has been changed into “After experimentation, the specimens were manufactured using a wire-cutting machine, and set with the cold inserts to sand conveniently for observation the microscopic structures. The microstructure was characterized through an optical microscope (OM), a scanning electron microscope (SEM), and an electron back scatter diffraction (EBSD). Some specimens for the observations of OM and SEM were corroded with ferric chloride aqueous solution, the ratio of which is HCl solution: water: FeCl3 solid = 30 mL: 120 mL: 10 g. The corrosion time of each weld specimen is 5 - 10 seconds. And the others were prepared using the electrochemical polishing for the observation of EBSD.”

Reviewer 2 Report
Revi
Review report
The article named “A New Test Method for Evaluation of Solidification Cracking Susceptibility of Stainless Steel during Laser Welding” discusses a new test method named “Trapezoidal hot” cracking test developed to evaluate the solidification cracking susceptibility of stainless-steel during laser welding. The solidification cracking, of SUS310, SUS316 and SUS304 stainless steels, was measured experimentally. This article indeed has a wide range of scientific information, novelty and conveys a piece of deterministic information. However, I have decided to assist authors in improving the quality of this manuscript for the final acceptance. Please find some comments from my side, which are necessary to improve the quality of the manuscript. The purpose of these comments is not the criticism but to improve the quality of this submission.
Regards,
The reviewer.
- Comments on the introduction section
Dear authors, when we quote the literature, we usually use the surnames, not initials of the first name such as “Shinozaki et al.”, not “Shinozaki, K. et al.” Please correct this mistake all over the manuscript. For your reference, please refer to “https://www.scribbr.com/apa-style/in-text-citation/#:~:text=Only%20the%20final%20name%20in,et%20al.%2C%202019).”
You have already mentioned the purpose of this study, which is really beneficial for the reader. But, please indicate the novelty of this work, with “purpose”, as it is necessary to attract the readers.
- Results and discussions
Please redefine the statement, line 235, Section 3.4. “With the increase of welding speed, the welding cooling rate increases, causing grain refinement.” This statement is not clear. Please explain in deep.
- English corrections
Line 15: A new test method named Trapezoidal hot cracking test was developed… should be… A new test method named “Trapezoidal hot” cracking test was developed…
Line 16: The new test method was used to obtain the solidification cracking directly and the solidification cracking susceptibility could be evaluated by the solidification cracking rate which was the ratio of the solidification cracking length to the weld bead length under the certain condition… should be… The new test method was used to obtain the solidification cracking directly, and the solidification cracking susceptibility could be evaluated by the solidification cracking rate, defined as the ratio of the solidification cracking length to the weld bead length under the certain condition.
Line 19: The results show that with the increasing of the solidification cracking rate, the solidification cracking susceptibility of SUS310 stainless steel was much higher than those of SUS316 and SUS304 stainless steels during laser welding at welding speed of 1.0 m/min because fully austenite appeared in the weld joint of the former steel, while the others were ferrite and austenitic mixed structure during solidification… should be… The results show that with the increasing of the solidification cracking rate, the solidification cracking susceptibility of SUS310 stainless steel was much higher than those of SUS316 and SUS304 stainless steels during laser welding at a welding speed of 1.0 m/min because fully austenite appeared in the weld joint of the former steel, while the others were ferrite and austenitic mixed structure during solidification.
Line 23: In addition, with an increase in welding speed from 1.0 m/min to 2.0 m/min during laser welding, the solidification cracking susceptibility of SUS310 stainless steel decreased slightly, however there was a tendency of increasing in the solidification cracking susceptibility of SUS304 stainless steel due to the decreasing of the amount of ferrite under a higher cooling rate… should be… Besides, with an increase in welding speed from 1.0 m/min to 2.0 m/min during laser welding, the solidification cracking susceptibility of SUS310 stainless steel decreased slightly; however, there was a tendency of increasing in the solidification cracking susceptibility of SUS304 stainless steel due to the decreasing of the amount of ferrite under a higher cooling rate.
Line 37: Please eliminate the “comma” after the word “composition” as it is a punctuation mistake.
Line 38: However, there are rare research on the solidification cracking susceptibility during laser welding… should read as… However, there is rare research on the solidification cracking susceptibility during laser welding.
Line 41: One is external restraint test method, such as Varestraint test [4-6].. should be… One is an external restraint test method, such as Varestraint test [4-6].
Line 45: Please eliminate the “comma” after the word “while” as it is a punctuation mistake.
Line 50: such as the Varestraint test machine, high speed camera.. should read as…such as the Varestraint test machine, high-speed camera.
Line 67: The used materials with the thickness of 2 mm are SUS310, SUS316 and SUS304 austenitic stainless steels.. should read as... The used materials with a thickness of 2 mm are SUS310, SUS316, and SUS304 austenitic stainless steels.
Line 67-70: The used materials with the thickness of 2 mm are SUS310, SUS316 and SUS304 austenitic stainless steels. And, SUS310 stainless steel was firstly employed to develop the new test method to evaluate the solidification cracking susceptibility. The chemical composition of these stainless steels are shown in Table 1… should read as… SUS310, SUS316 and SUS304 austenitic stainless steels having a thickness of 2 mm, were used for experimentations. Initially, SUS310 stainless steel was employed to develop the new test method for the evaluation of solidification cracking susceptibility. The chemical composition of these stainless steels is shown in Table 1.
Line 74: Table 2 shows the laser welding conditions. Figure 1 exhibits the experimental equipment and method. A fiber laser was employed as a heat source and the wavelength is 1064 nm. The welding speeds are 1.0 m/min, 1.5 m/min and 2.0 m/min and the corresponding laser powers are 1.8 kW, 2.3 kW and 2.8 kW respectively in order to obtain the full penetration…should read as…Table 2 shows the laser welding conditions. Figure 1 shows the experimental equipment and the respective method. A fiber laser having 1064 nm wavelength was employed as a heat source. To obtain the full penetration, the welding speeds equivalent to 1.0, 1.5 and 2.0 m/min and the corresponding laser powers equal to1.8 kW, 2.3 kW and 2.8 kW, respectively, were utilized.
Line 83: After test, the specimens were manufactured using a wire-cutting machine and the metallographic specimens were corroded with ferric chloride aqueous solution… should read as… After experimentation, the specimens were manufactured using a wire-cutting machine, and the metallographic specimens were corroded with ferric chloride aqueous solution.
Line 96: Figure 2 shows the schematic image of the specimen of the new test method to evaluate the solidification cracking susceptibility during laser welding… should read as… Figure 2 shows the schematic image of the specimen for the new test method to evaluate the solidification cracking susceptibility during laser welding.
Line 97: The specimen geometry of this new test method is like a trapezoid, so it is called as “Trapezoidal hot cracking test”, as shown in Figure 2a. While, the key to develop this trapezoidal test is to decide the values of WS and WF who are the beginning and the end edges of the welding respectively. In order to ensure that the full penetration weld bead can be obtained during laser welding, especially at the beginning, the wedge groove of 1 mm × 5 mm was be machined out at the beginning edge of the specimen, as shown in Figure 2b… should read as… The specimen geometry for this new test method is the same as a trapezoid, so it is called as “Trapezoidal hot cracking test,” as shown in Figure 2a. The key behind this trapezoidal test is to decide the values of WS and WF, defined as the starting and final edges of the welding process, respectively. To ensure that the full penetration weld bead can be obtained, during laser welding, especially at the beginning, the wedge groove of 1 mm × 5 mm was machined out at the starting edge of the specimen, as shown in Figure 2b.
Line 105: The Trapezoidal hot cracking test belongs to the self-restraint test method which normally includes three steps, initiation, propagation and ceasing of the solidification cracking, respectively, to evaluate the solidification cracking susceptibility… should read as…The Trapezoidal hot cracking test belongs to the self-restraint test method, which normally includes three steps, initiation, propagation, and ceasing of the solidification cracking, respectively, to evaluate the solidification cracking susceptibility.
Line 107: Thus, it is necessary to make clear the effect of the each step in the case of the Trapezoidal hot cracking test… should read as… Thus, it is necessary to make clear the effect of each step in the case of the Trapezoidal hot cracking test.
Line 108: Importantly, the Trapezoidal hot cracking test wouldn’t just need to initiate the solidification cracking during welding, but also need to cease the cracking before the end of the welding, because it is impossible to evaluate and compare the susceptibility if there is no crack or the crack appears throughout the weld bead during each test. For this reason, the values of WS and WF play an important role in initiating, propagating and ceasing of the solidification cracking, respectively… should read as… Importantly, Trapezoidal hot cracking test wouldn’t just need to initiate the solidification cracking during welding, but also need to cease the cracking before the end of the welding, because it is impossible to evaluate and compare the susceptibility if there is no crack, or the crack appears throughout the weld bead during each test. For this reason, the values of WS and WF play an important role in initiating, propagating, and ceasing of the solidification cracking, respectively.
Line 114: During welding, the relatively higher weld stress will occur at the rear of the molten pool perpendicular to the direction of temperature gradient as a result of thermal expansion producing along the centerline of the weld bead… should read as… During welding, the relatively higher weld stress will occur at the rear of the melt pool perpendicular to the direction of the temperature gradient as a result of thermal expansion producing along with the centerline of the weld bead.
Line 118: If WS is too narrow, the solidification cracking will not initiate because of the relatively less deflection force as a result of lacking of temperature gradient… should read as… If WS is too narrow, the solidification cracking will not initiate because of the relatively less deflection force as a result of lacking of the temperature gradient.
Line 124: With the increasing of the specimen width along welding direction, the restraint will gradually increase… should read as… With the increase of the specimen width along the welding direction, the restraint will gradually increase.
Line 130: Figure 3 shows the method to obtain the suitable value of WS. The size of the specimen is 100 mm ×100 mm × 2 mm, firstly. The distance from the specimen edge is defined as (1/2)WS which is the beginning position of the welding, as shown in Figure 3a. Figure 4 shows the macro profile of the solidification cracking under the different values of (1/2)WS during laser welding at welding speed of 1.0 m/min. Figure 5 shows the solidification cracking length under the different values of (1/2)WS during laser welding at welding speed of 1.0 m/min corresponding to those of Figure 4. When the value of (1/2)WS was less than 6 mm, the solidification cracking appeared and the length was more than 80 mm at both the front and backside of the specimen. While, there was no crack under the value of (1/2)WS more than 7 mm, as shown in Figures 4 and 5. According to the above principle, the value of WS could be set 12 mm in order to initiate crack at the beginning of the welding. … should read as… Figure 3 shows the method to obtain a suitable value of WS. The size of the specimen is 100 mm ×100 mm × 2 mm, initially. The distance from the specimen edge is defined as (1/2)WS, which is the beginning position of the welding, as shown in Figure 3a. Figure 4 shows the macro profile of the solidification cracking under the different values of (1/2)WS during laser welding at a welding speed of 1.0 m/min. Figure 5 shows the solidification cracking length under the different values of (1/2)WS during laser welding at a welding speed of 1.0 m/min corresponding to those of Figure 4. When the value of (1/2)WS was less than 6 mm, the solidification cracking appeared, and the length was more than 80 mm at both the front and backside of the specimen. While, there was no crack under the value of (1/2)WS more than 7 mm, as shown in Figures 4 and 5. According to the above principle, the value of WS could be set 12 mm to initiate crack at the beginning of the welding.
Line 142: Figure 4. The macro profile of the solidification cracking of SUS310 stainless steel under the different values of (1/2)WS during laser welding at welding speed of 1.0 m/min… should read as… Figure 4. The macro profile of the solidification cracking of SUS310 stainless steel under the different values of (1/2)WS during laser welding at a welding speed of 1.0 m/min.
Line 145: Effect of (1/2)WS on the solidification cracking length during laser welding at welding speed of 1.0 m/min… should read as…: Effect of (1/2)WS on the solidification cracking length during laser welding at a welding speed of 1.0 m/min.
Line 147: Then, the value of WF was assumed as 20 mm, 30 mm, 40 mm and 50 mm, respectively. Figure 6 shows the macroscopic appearance of the solidification cracking of SUS310 stainless steel under WS 149 of 12 mm and the different values of WF during laser welding at welding speed of 1.0 m/min… should read as… Then, the value of WF was assumed as 20 mm, 30 mm, 40 mm, and 50 mm, respectively. Figure 6 shows the macroscopic appearance of the solidification cracking of SUS310 stainless steel under WS 149 of 12 mm and the different values of WF during laser welding at a welding speed of 1.0 m/min.
Line 154: However, under the certain condition, the solidification cracking susceptibility can be characterized by the cracking length… should read as… However, under a certain condition, the solidification cracking susceptibility can be characterized by the cracking length.
Line 162: Figure 6. Macroscopic appearance of the solidification cracking of SUS310 stainless steel under WS of 12 mm and the different values of WF during laser welding at welding speed of 1.0 m/min… should read as… Figure 6. Macroscopic appearance of the solidification cracking of SUS310 stainless steel under WS of 12 mm and the different values of WF during laser welding at a welding speed of 1.0 m/min.
Line 165: Figure 7. Effect of WF on the solidification cracking length under WS of 12 mm during laser welding at welding speed of 1.0 m/min… should read as… Figure 7. Effect of WF on the solidification cracking length under WS of 12 mm during laser welding at a welding speed of 1.0 m/min.
Line 169: Figure 8 shows the effect of the chemical components (different stainless steels) on the
solidification cracking rate during laser welding as welding speed of 1.0 m/min. The sequence of the solidification cracking rate under WS of 12 mm and WF of 40 mm is as follow: SUS310 > SUS316 > SUS304. Especially, the solidification cracking rate of SUS310 stainless steel was much higher than the others… should read as… Figure 8 shows the effect of the chemical components (different stainless steels) on the solidification cracking rate during laser welding at a welding speed of 1.0 m/min. The sequence of the solidification cracking rate under WS of 12 mm and WF of 40 mm is as follows: SUS310 > SUS316 > SUS304. Especially, the solidification cracking rate of SUS310 stainless steel was much higher than the others.
Line 175: Figure 8. Effect of the chemical components on the solidification cracking rate during laser welding 176 at welding speed of 1.0 m/min…should read as… Figure 8. Effect of the chemical components on the solidification cracking rate during laser welding 176 at the welding speed of 1.0 m/min.
Line 180: The austenite structure and some lath ferrite structure could be observed from the microstructure images of SUS316 and SUS304 stainless steels welding, and further the amount of ferrite structure in SUS304 stainless steel seemed to be more than that of SUS316 stainless steel, as shown in Figure 9b, c, e and f… should read as…The austenite some lath ferrite structures could be observed from the microstructure images of SUS316 and SUS304 stainless steels welding; further, the amount of ferrite structure in SUS304 stainless steel seemed to be more than that of SUS316 stainless steel, as shown in Figures 9b, c, e, and f.
Line 184: According to the welding metallurgy of stainless steel and previous researches, the microstructures of the weld bead are depended on the solidification mode[26], while the value of 186 Creq/Nieq can determine the solidification mode of the welding… should read as…According to the welding metallurgy of stainless steel and previous researches, the microstructures of the weld bead are dependent on the solidification mode [26]. At the same time, the value of Creq/Nieq can determine the solidification mode of the welding.
Line 190: Please change the word “A mode” with “A-mode.”
Line 191: When the value of Creq/Nieq is between 1.25 and 1.48, the initial phase is austenite and then δ-ferrite is formed by reaction, so it is called AF mode… should read as… When the value of Creq/Nieq is between 1.25 and 1.48, the initial phase is austenite, and then δ-ferrite is formed by reaction, so it is called AF mode.
Line 192: Since the ferrite forming elements, such as Cr, can segregate in the subgrain boundary so as to promote the formation of ferrite, the cracking susceptibility could be reduced to a certain extent… should read as… Since the ferrite forming elements, such as Cr, can segregate in the subgrain boundary to promote the formation of ferrite, the cracking susceptibility could be reduced to a certain extent.
Line 196: Because the primary ferrite can solute higher impurities, like S and P, that is beneficial to restricting the partitioning of these elements to the interdendritic regions causing a good resistance to the solidification cracking. If the value of Creq/Nieq is greater than 1.95, it forms the fully ferrite during the solidification [27]… should read as… Because the primary ferrite can solute higher impurities, like S and P, that is beneficial to restricting the partitioning of these elements to the interdendritic regions, causing a good resistance to the solidification cracking. If the value of Creq/Nieq is greater than 1.95, it forms the full ferrite during the solidification [27].
Line 200: Through calculation, the values of Creq/Nieq in SUS310, SUS316 and SUS304 stainless steels were 201 1.11, 1.53, and 1.91, respectively… should read as… Through calculation, the values of Creq/Nieq in SUS310, SUS316, and SUS304 stainless steels were 201 1.11, 1.53, and 1.91, respectively.
Line 202: Obviously, the fully austenitic structure contributed to the more susceptibility compared with ferrite and austenitic mixed structure, as shown in Figure 9… should be corrected as… The fully austenitic structure contributed to the more susceptibility compared with ferrite and austenitic mixed structure, as shown in Figure 9.
Line 214: OM and SEM images of the microstructure of the cross-section of weld joint during laser welding at welding speed of 1.0 m/min… should read as…OM and SEM images of the microstructure of the cross-section of the weld joint during laser welding at a welding speed of 1.0 m/min.
Line 218: Figure 10 shows the effect of welding speed on the solidification cracking rate of SUS310 and SUS304 stainless steels during laser welding using the Trapezoidal hot cracking test with WS of 12 mm. For SUS310 stainless steel, the solidification cracking rate was done under WF of 40 mm during laser welding at different welding speeds from 1.0 m/min to 2.0 m/min, since the cracking rate was already the relatively high during laser welding at welding speed of 1.0 m/min under WF of 20 mm. In addition, due to the relatively low solidification cracking rate in SUS304 stainless steel compared with that of SUS310 stainless steel, the value of WF was set 20 mm in order to measure and compare the cracking rate conveniently. The results showed that with the increase of laser welding speed, the solidification cracking rate of SUS310 stainless steel tended to decrease gradually, that is, the cracking susceptibility dropped slowly in the case of A mode solidification. In contrast, there was a tendency of increasing in the solidification cracking rate of SUS304 stainless steel with the increase of welding speed, that is, the cracking susceptibility augmented in the case of FA mode solidification…. Should be corrected as… Figure 10 shows the effect of welding speed on the solidification cracking rate of SUS310 and SUS304 stainless steels during laser welding using the Trapezoidal hot cracking test with WS of 12 mm. For SUS310 stainless steel, the solidification cracking rate was done under WF of 40 mm during laser welding at different welding speeds from 1.0 m/min to 2.0 m/min, since the cracking rate was already the relatively high during laser welding at a welding speed of 1.0 m/min under WF of 20 mm. Besides, due to the relatively low solidification cracking rate in SUS304 stainless steel compared with that of SUS310 stainless steel, the value of WF was set 20 mm to measure and compare the cracking rate conveniently. The results showed that with the increase of laser welding speed, the solidification cracking rate of SUS310 stainless steel tended to decrease gradually, that is, the cracking susceptibility dropped slowly in the case of A mode solidification. In contrast, there was a tendency of increasing in the solidification cracking rate of SUS304 stainless steel with the increase of welding speed, that is, the cracking susceptibility augmented in the case of FA mode solidification.
Line 231: The grains in SUS310 stainless steel had a certain degree of refinement during the increase of welding speed, as shown in Figure 11a, b. While, for SUS304 stainless steel during laser welding, the amount of δ-ferrite in the microstructure was obviously less at welding speed of 2.0 m/min than those at 1.0 m/min, as shown in Figure 11c, d… should read as… The grains in SUS310 stainless steel had a certain degree of refinement during the increase of welding speed, as shown in Figure 11a, b. While, for SUS304 stainless steel during laser welding, the amount of δ-ferrite in the microstructure was less at a welding speed of 2.0 m/min than those at 1.0 m/min, as shown in Figure 11c, d.
Line 235: With the increase of welding speed, the welding cooling rate increases, causing grain refinement. During solidification, the grain refinement can not only promote the refill and healing of liquid phase more effectively, but also increase the number of grain boundaries, further preventing harmful lowmelting-point compound from segregating on grain boundaries [29]… should be… With the increase of welding speed, the welding cooling rate increases, causing grain refinement. During solidification, the grain refinement can not only promote the refill and healing of liquid phase more effectively but also increase the number of grain boundaries, further preventing harmful low melting-point compound from segregating on grain boundaries.
Line 248: Figure 10. Effect of the welding speed on the solidification cracking rate during laser welding… should read as… Effect of the welding speed on the solidification cracking rate during laser welding.
Line 251: OM image of the microstructure of the cross-section of weld joint… should read as… OM image of the microstructure of the cross-section of the weld joint.
Line 258: The new test method called the Trapezoidal hot cracking test was developed to evaluate the solidification cracking susceptibility using the solidification cracking rate which was the ratio of the solidification cracking length to the weld bead length under the suitable values of WS and WF… should be…The new test method called the Trapezoidal hot cracking test was developed to evaluate the solidification cracking susceptibility using the solidification cracking rate which was the ratio of the solidification cracking length to the weld bead length under the suitable values of WS and WF.
Line 261: The sequence of the solidification cracking susceptibility was as follow: SUS310 > SUS316 > SUS304 during laser welding at welding speed of 1.0 m/min… should read as… The sequence of the solidification cracking susceptibility was as follow: SUS310 > SUS316 > SUS304 during laser welding at a welding speed of 1.0 m/min.
Line 267: This is due to the grain refinement for the former and a decrease in the amount of ferrite for the latter…. should be… It is due to the grain refinement for the former and a decrease in the amount of ferrite for the latter.

Author Response
Response to Reviewer 2 Comments
Thank you very much for your comments. The authors also upload an attachment.
Point 1: Dear authors, when we quote the literature, we usually use the surnames, not initials of the first name such as “Shinozaki et al.”, not “Shinozaki, K. et al.” Please correct this mistake all over the manuscript.
Response 1: Thank you very much for your careful comments. We are very sorry for our negligence of these mistakes for the authors’ name. And the detailed revision is as follow:
In line 46, “Shinozaki, K.[17-22] et al. developed U type hot cracking test with laser welding and laser Trans-Varestraint test[17-22] to evaluate solidification cracking susceptibility of stainless steel quantitatively.” has been changed into “Shinozaki[17-22] et al. developed U type hot cracking test with laser welding and laser Trans-Varestraint test[17-22] to evaluate solidification cracking susceptibility of stainless steel quantitatively.”
In line 51-53, “The solidification cracking susceptibility of aluminium alloy was evaluated quantitatively using the Houldcroft test during laser welding with filler wire by Zhai, Y.[23].” has been changed into “The solidification cracking susceptibility of aluminum alloy was evaluated quantitatively using the Houldcroft test during laser welding with filler wire by Zhai[23].”
In line 57-58, “Fukuhisa, M.[24] et al. developed a new test method to evaluate the solidification cracking susceptibility of not only aluminum alloy but also stainless steel during electron beam welding.” has been changed into “Fukuhisa[24] et al. developed a new test method to evaluate the solidification cracking susceptibility of not only aluminum alloy but also stainless steel during electron beam welding.”
In line 155-156, “Reddy, G.[25] et al. indicated that the solidification cracking susceptibility could be evaluated using the cracking length of the specimen after welding.” has been changed into “Reddy[25] et al. indicated that the solidification cracking susceptibility could be evaluated by using the cracking length of the specimen after welding.
In line 238-241, “Moreover, Agarwal, G.[30] et al pointed out that during SUS310 stainless steel laser welding, with the increase of welding speed, the solidification segregation was inhibited and the brittleness temperature range was reduced, resulting in a decrease in the solidification cracking susceptibility.” has been changed into “Moreover, Agarwal[31] et al pointed out that during SUS310 stainless steel laser welding, with the increase of welding speed, the solidification segregation was inhibited and the brittleness temperature range was reduced, resulting in a decrease in the solidification cracking susceptibility.”
Point 2: You have already mentioned the purpose of this study, which is really beneficial for the reader. But, please indicate the novelty of this work, with “purpose”, as it is necessary to attract the readers.
Response 2: It is kind of you for your valuable comments on our manuscript. We have already the corresponding contents in order to further show the advantages of our new test method and attract the readers. The detailed revision is as follow:
In line 62, “The purpose of this research is to develop a new test method to evaluate the effect of the chemical composition on the solidification cracking susceptibility of stainless steel during laser welding at different welding speeds.” has been changed into “Certainly, the advantages of the new test method would be obvious compared with the previous methods. Firstly, it should have a low requirement on the equipment, that is, it is better to only need a heat source and some simple specimen to evaluate the solidification cracking susceptibility. Besides, the heat source of this new test method could be changed easily and applicable not only for a laser, but also for the others, like GTAW. Thirdly, the new test method may evaluate the solidification cracking susceptibility of kinds of materials, such as stainless steel or light alloy, etc. In a word, the developed new test method should have the characteristic of the simple equipment, widely application, low cost and reliable. Therefore, the purpose of this research is to develop a new test method owning the advantages mentioned above to evaluate the effect of the chemical composition on the solidification cracking susceptibility of stainless steel during laser welding at different welding speeds.”
Point 3: Please redefine the statement, line 235, Section 3.4. “With the increase of welding speed, the welding cooling rate increases, causing grain refinement.” This statement is not clear. Please explain in deep.
Response 3: Thank you again for your valuable suggestions.
The detailed revision is as follow:
In line 235, “With the increase of welding speed, the welding cooling rate increases, causing grain refinement.” has been changed into “Theoretically, when the power is fixed during laser welding, the welding heat input, calculated using the power divided by the welding speed, will decrease with the increase of welding speed, resulting in the appearance of a relatively higher cooling rate, and further causing grain refinement[29]. While, the heat inputs were 108 J/mm for a welding speed of 1.0 m/min, 92 J/mm for a welding speed of 1.5 m/min and 84 J/mm for a welding speed of 2.0 m/min through the calculation, respectively. Thus, the cooling rate tended to increase with increasing welding speed from 1.0 m/min to 2.0 m/min in this experiment during laser welding, inducing to the phenomenon that the grain size slowed down.”
Point 4: In line 15, A new test method named Trapezoidal hot cracking test was developed… should be… A new test method named “Trapezoidal hot” cracking test was developed…
Response 4: The detailed revision is as follow:
In line 15,“A new test method named Trapezoidal hot cracking test was developed to evaluate solidification cracking susceptibility of stainless steel during laser welding.” has been changed into “A new test method named “Trapezoidal hot” cracking test was developed to evaluate solidification cracking susceptibility of stainless steel during laser welding.”
Point 5: In line 16, The new test method was used to obtain the solidification cracking directly and the solidification cracking susceptibility could be evaluated by the solidification cracking rate which was the ratio of the solidification cracking length to the weld bead length under the certain condition… should be… The new test method was used to obtain the solidification cracking directly, and the solidification cracking susceptibility could be evaluated by the solidification cracking rate, defined as the ratio of the solidification cracking length to the weld bead length under the certain condition.
Response 5: The detailed revision is as follow:
In line 16, “The new test method was used to obtain the solidification cracking directly and the solidification cracking susceptibility could be evaluated by the solidification cracking rate which was the ratio of the solidification cracking length to the weld bead length under the certain condition.” has been changed into “The new test method was used to obtain the solidification cracking directly, and the solidification cracking susceptibility could be evaluated by the solidification cracking rate, defined as the ratio of the solidification cracking length to the weld bead length under the certain condition.”
Point 6: Line 19: The results show that with the increasing of the solidification cracking rate, the solidification cracking susceptibility of SUS310 stainless steel was much higher than those of SUS316 and SUS304 stainless steels during laser welding at welding speed of 1.0 m/min because fully austenite appeared in the weld joint of the former steel, while the others were ferrite and austenitic mixed structure during solidification… should be… The results show that with the increasing of the solidification cracking rate, the solidification cracking susceptibility of SUS310 stainless steel was much higher than those of SUS316 and SUS304 stainless steels during laser welding at a welding speed of 1.0 m/min because fully austenite appeared in the weld joint of the former steel, while the others were ferrite and austenitic mixed structure during solidification.
Response 6: The detailed revision is as follow:
In line 19, “The results show that with the increasing of the solidification cracking rate, the solidification cracking susceptibility of SUS310 stainless steel was much higher than those of SUS316 and SUS304 stainless steels during laser welding at welding speed of 1.0 m/min because fully austenite appeared in the weld joint of the former steel, while the others were ferrite and austenitic mixed structure during solidification.” has been changed into “The results show that with the increasing of the solidification cracking rate, the solidification cracking susceptibility of SUS310 stainless steel was much higher than those of SUS316 and SUS304 stainless steels during laser welding at a welding speed of 1.0 m/min because fully austenite appeared in the weld joint of the former steel, while the others were ferrite and austenitic mixed structure during solidification.”
Point 7: Line 23: In addition, with an increase in welding speed from 1.0 m/min to 2.0 m/min during laser welding, the solidification cracking susceptibility of SUS310 stainless steel decreased slightly, however there was a tendency of increasing in the solidification cracking susceptibility of SUS304 stainless steel due to the decreasing of the amount of ferrite under a higher cooling rate… should be… Besides, with an increase in welding speed from 1.0 m/min to 2.0 m/min during laser welding, the solidification cracking susceptibility of SUS310 stainless steel decreased slightly; however, there was a tendency of increasing in the solidification cracking susceptibility of SUS304 stainless steel due to the decreasing of the amount of ferrite under a higher cooling rate.
Response 7: The detailed revision is as follow:
In line 23, “In addition, with an increase in welding speed from 1.0 m/min to 2.0 m/min during laser welding, the solidification cracking susceptibility of SUS310 stainless steel decreased slightly, however there was a tendency of increasing in the solidification cracking susceptibility of SUS304 stainless steel due to the decreasing of the amount of ferrite under a higher cooling rate.” has been changed into “Besides, with an increase in welding speed from 1.0 m/min to 2.0 m/min during laser welding, the solidification cracking susceptibility of SUS310 stainless steel decreased slightly; however, there was a tendency of increasing in the solidification cracking susceptibility of SUS304 stainless steel due to the decreasing of the amount of ferrite under a higher cooling rate.”
Point 8: Line 37: Please eliminate the “comma” after the word “composition” as it is a punctuation mistake.
Response 8: The detailed revision is as follow:
In line 37, “While, solidification cracking susceptibility could be changed during laser welding with relatively higher welding speed compared with that of gas tungsten arc welding (GTAW), since the susceptibility can be affected by solidification behaviour which depends on not only chemical composition, but also welding conditions, such as the welding speed considered as a very important factor.” has been changed into “While, solidification cracking susceptibility could be changed during laser welding with relatively higher welding speed compared with that of gas tungsten arc welding (GTAW), since the susceptibility can be affected by solidification behaviour which depends on not only chemical composition but also welding conditions, such as the welding speed considered as a very important factor.”
Point 9: Line 38: However, there are rare research on the solidification cracking susceptibility during laser welding… should read as… However, there is rare research on the solidification cracking susceptibility during laser welding.
Response 9: The detailed revision is as follow:
In line 38, “However, there are rare research on the solidification cracking susceptibility during laser welding.” has been changed into “However, there is rare research on the solidification cracking susceptibility during laser welding.”
Point 10: Line 41: One is external restraint test method, such as Varestraint test [4-6]… should be… One is an external restraint test method, such as Varestraint test [4-6].
Response 10: The detailed revision is as follow:
In line 41, “One is external restraint test method, such as Varestraint test[4-6], Sigmajig test[7,8], Rapid tensile test[9,10], etc.” has been changed into “One is an external restraint test method, such as Varestraint test [4-6], Sigmajig test[7,8], Rapid tensile test[9,10], etc.”
Point 11: Line 45: Please eliminate the “comma” after the word “while” as it is a punctuation mistake. Response 11: The detailed revision is as follow:
In line 45, “While, there are few test methods to evaluate the susceptibility during laser welding.” has been changed into “While there are few test methods to evaluate the susceptibility during laser welding.”
Point 12: Line 50: such as the Varestraint test machine, high speed camera… should read as…such as the Varestraint test machine, high-speed camera.
Response 12: The detailed revision is as follow:
In line 50, “However, these methods are hard for widespread use, since it is difficult to measure temperature accurately during laser welding at high welding speed and also it needs expensive equipment, such as the Varestraint test machine, high speed camera, and high precision temperature measurement device and so on.” has been changed into “However, these methods are hard for widespread use, since it is difficult to measure temperature accurately during laser welding at high welding speed and also it needs expensive equipment, such as the Varestraint test machine, high-speed camera, and high precision temperature measurement device and so on.”
Point 13: Line 67: The used materials with the thickness of 2 mm are SUS310, SUS316 and SUS304 austenitic stainless steels…should read as... The used materials with a thickness of 2 mm are SUS310, SUS316, and SUS304 austenitic stainless steels.
Response 13: The detailed revision is as follow:
In line 67, “The used materials with the thickness of 2 mm are SUS310, SUS316 and SUS304 austenitic stainless steels.” has been changed into “The used materials with a thickness of 2 mm are SUS310, SUS316, and SUS304 austenitic stainless steels.”
Point 14: Line 67-70: The used materials with the thickness of 2 mm are SUS310, SUS316 and SUS304 austenitic stainless steels. And, SUS310 stainless steel was firstly employed to develop the new test method to evaluate the solidification cracking susceptibility. The chemical composition of these stainless steels are shown in Table 1… should read as… SUS310, SUS316 and SUS304 austenitic stainless steels having a thickness of 2 mm, were used for experimentations. Initially, SUS310 stainless steel was employed to develop the new test method for the evaluation of solidification cracking susceptibility. The chemical composition of these stainless steels is shown in Table 1.
Response 14: The detailed revision is as follow:
In line 67-70, “The used materials with the thickness of 2 mm are SUS310, SUS316 and SUS304 austenitic stainless steels. And, SUS310 stainless steel was firstly employed to develop the new test method to evaluate the solidification cracking susceptibility. The chemical composition of these stainless steels are shown in Table 1.” has been changed into “SUS310, SUS316 and SUS304 austenitic stainless steels having a thickness of 2 mm, were used for experimentations. Initially, SUS310 stainless steel was employed to develop the new test method for the evaluation of solidification cracking susceptibility. The chemical composition of these stainless steels is shown in Table 1.”
Point 15: Line 74: Table 2 shows the laser welding conditions. Figure 1 exhibits the experimental equipment and method. A fiber laser was employed as a heat source and the wavelength is 1064 nm. The welding speeds are 1.0 m/min, 1.5 m/min and 2.0 m/min and the corresponding laser powers are 1.8 kW, 2.3 kW and 2.8 kW respectively in order to obtain the full penetration…should read as…Table 2 shows the laser welding conditions. Figure 1 shows the experimental equipment and the respective method. A fiber laser having 1064 nm wavelength was employed as a heat source. To obtain the full penetration, the welding speeds equivalent to 1.0, 1.5 and 2.0 m/min and the corresponding laser powers equal to1.8 kW, 2.3 kW and 2.8 kW, respectively, were utilized.
Response 15: The detailed revision is as follow:
In line 74, “Table 2 shows the laser welding conditions. Figure 1 shows the experimental equipment and method. A fiber laser was employed as a heat source and the wavelength is 1064 nm. The welding speeds are 1.0 m/min, 1.5 m/min and 2.0 m/min and the corresponding laser powers are 1.8 kW, 2.3 kW and 2.8 kW respectively in order to obtain the full penetration.” has been changed into “Table 2 shows the laser welding conditions. Figure 1 shows the experimental equipment and the respective method. A fiber laser having 1064 nm wavelength was employed as a heat source. To obtain the full penetration, the welding speeds equivalent to 1.0, 1.5 and 2.0 m/min and the corresponding laser powers equal to1.8 kW, 2.3 kW and 2.8 kW, respectively, were utilized.”
Point 16: Line 83: After test, the specimens were manufactured using a wire-cutting machine and the metallographic specimens were corroded with ferric chloride aqueous solution… should read as… After experimentation, the specimens were manufactured using a wire-cutting machine, and the metallographic specimens were corroded with ferric chloride aqueous solution.
Response 16: The detailed revision is as follow:
In line 83, “After test, the specimens were manufactured using a wire-cutting machine and the metallographic specimens were corroded with ferric chloride aqueous solution.” has been changed into “After experimentation, the specimens were manufactured using a wire-cutting machine, and the metallographic specimens were corroded with ferric chloride aqueous solution.”
Point 17: Line 96: Figure 2 shows the schematic image of the specimen of the new test method to evaluate the solidification cracking susceptibility during laser welding… should read as… Figure 2 shows the schematic image of the specimen for the new test method to evaluate the solidification cracking susceptibility during laser welding.
Response 17: The detailed revision is as follow:
In line 96, “Figure 2 shows the schematic image of the specimen of the new test method to evaluate the solidification cracking susceptibility during laser welding.” has been changed into “Figure 2 shows the schematic image of the specimen for the new test method to evaluate the solidification cracking susceptibility during laser welding.”
Point 18: Line 97: The specimen geometry of this new test method is like a trapezoid, so it is called as “Trapezoidal hot cracking test”, as shown in Figure 2a. While, the key to develop this trapezoidal test is to decide the values of WS and WF who are the beginning and the end edges of the welding respectively. In order to ensure that the full penetration weld bead can be obtained during laser welding, especially at the beginning, the wedge groove of 1 mm × 5 mm was be machined out at the beginning edge of the specimen, as shown in Figure 2b… should read as… The specimen geometry for this new test method is the same as a trapezoid, so it is called as “Trapezoidal hot cracking test,” as shown in Figure 2a. The key behind this trapezoidal test is to decide the values of WS and WF, defined as the starting and final edges of the welding process, respectively. To ensure that the full penetration weld bead can be obtained, during laser welding, especially at the beginning, the wedge groove of 1 mm × 5 mm was machined out at the starting edge of the specimen, as shown in Figure 2b.
Response 18: The detailed revision is as follow:
In line 97, “The specimen geometry of this new test method is like a trapezoid, so it is called as “Trapezoidal hot cracking test”, as shown in Figure 2a. While, the key to develop this trapezoidal test is to decide the values of WS and WF who are the beginning and the end edges of the welding respectively. In order to ensure that the full penetration weld bead can be obtained during laser welding, especially at the beginning, the wedge groove of 1 mm × 5 mm was be machined out at the beginning edge of the specimen, as shown in Figure 2b.” has been changed into “The specimen geometry for this new test method is the same as a trapezoid, so it is called as “Trapezoidal hot” cracking test, as shown in Figure 2a. The key behind this trapezoidal test is to decide the values of WS and WF, defined as the starting and final edges of the welding process, respectively. To ensure that the full penetration weld bead can be obtained, during laser welding, especially at the starting, the wedge groove of 1 mm × 5 mm was machined out at the starting edge of the specimen, as shown in Figure 2b.”
Point 19: Line 105: The Trapezoidal hot cracking test belongs to the self-restraint test method which normally includes three steps, initiation, propagation and ceasing of the solidification cracking, respectively, to evaluate the solidification cracking susceptibility… should read as…The Trapezoidal hot cracking test belongs to the self-restraint test method, which normally includes three steps, initiation, propagation, and ceasing of the solidification cracking, respectively, to evaluate the solidification cracking susceptibility.
Response 19: The detailed revision is as follow:
In line 105, “The Trapezoidal hot cracking test belongs to the self-restraint test method which normally includes three steps, initiation, propagation and ceasing of the solidification cracking, respectively, to evaluate the solidification cracking susceptibility.” has been changed into “The ‘Trapezoidal hot’ cracking test belongs to the self-restraint test method, which normally includes three steps, initiation, propagation, and ceasing of the solidification cracking, respectively, to evaluate the solidification cracking susceptibility.”
Point 20: Line 107: Thus, it is necessary to make clear the effect of the each step in the case of the Trapezoidal hot cracking test… should read as… Thus, it is necessary to make clear the effect of each step in the case of the Trapezoidal hot cracking test.
Response 20: The detailed revision is as follow:
In line 107, “Thus, it is necessary to make clear the effect of the each step in the case of the Trapezoidal hot cracking test.” has been changed into “Thus, it is necessary to clarify the effect of each step in the case of the ‘Trapezoidal hot’ cracking test.”
Point 21: Line 108: Importantly, the Trapezoidal hot cracking test wouldn’t just need to initiate the solidification cracking during welding, but also need to cease the cracking before the end of the welding, because it is impossible to evaluate and compare the susceptibility if there is no crack or the crack appears throughout the weld bead during each test. For this reason, the values of WS and WF play an important role in initiating, propagating and ceasing of the solidification cracking, respectively… should read as… Importantly, Trapezoidal hot cracking test wouldn’t just need to initiate the solidification cracking during welding, but also need to cease the cracking before the end of the welding, because it is impossible to evaluate and compare the susceptibility if there is no crack, or the crack appears throughout the weld bead during each test. For this reason, the values of WS and WF play an important role in initiating, propagating, and ceasing of the solidification cracking, respectively.
Response 21: The detailed revision is as follow:
In line 109-114, “Importantly, the Trapezoidal hot cracking test wouldn’t just need to initiate the solidification cracking during welding, but also need to cease the cracking before the end of the welding, because it is impossible to evaluate and compare the susceptibility if there is no crack or the crack appears throughout the weld bead during each test. For this reason, the values of WS and WF play an important role in initiating, propagating and ceasing of the solidification cracking, respectively.” has been changed into “Importantly, trapezoidal test would not just need to initiate the solidification cracking during welding, but also need to cease the cracking before the end of the welding, because it is impossible to evaluate and compare the susceptibility if there is no crack, or the crack appears throughout the weld bead during each test. For this reason, the values of WS and WF play an important role in initiating, propagating, and ceasing of the solidification cracking, respectively."
Point 22: Line 114: During welding, the relatively higher weld stress will occur at the rear of the molten pool perpendicular to the direction of temperature gradient as a result of thermal expansion producing along the centerline of the weld bead… should read as… During welding, the relatively higher weld stress will occur at the rear of the melt pool perpendicular to the direction of the temperature gradient as a result of thermal expansion producing along with the centerline of the weld bead.
Response 22: The detailed revision is as follow:
In line 114, “During welding, the relatively higher weld stress will occur at the rear of the molten pool perpendicular to the direction of temperature gradient as a result of thermal expansion producing along the centerline of the weld bead.” has been changed into “During welding, the relatively higher weld stress will occur at the rear of the melt pool perpendicular to the direction of the temperature gradient as a result of thermal expansion producing along with the centerline of the weld bead.”
Point 23: Line 118: If WS is too narrow, the solidification cracking will not initiate because of the relatively less deflection force as a result of lacking of temperature gradient… should read as… If WS is too narrow, the solidification cracking will not initiate because of the relatively less deflection force as a result of lacking of the temperature gradient.
Response 23: The detailed revision is as follow:
In line 118, “If WS is too narrow, the solidification cracking will not initiate because of the relatively less deflection force as a result of lacking of temperature gradient.” has been changed into “If WS is too narrow, the solidification cracking will not initiate because of the relatively less deflection force as a result of lacking of the temperature gradient.”
Point 24: Line 124: With the increasing of the specimen width along welding direction, the restraint will gradually increase… should read as… With the increase of the specimen width along the welding direction, the restraint will gradually increase.
Response 24: The detailed revision is as follow:
In line 124, “With the increasing of the specimen width along welding direction, the restraint will gradually increase.” has been changed into “With the increase of the specimen width along the welding direction, the restraint will gradually increase.”
Point 25: Line 130: Figure 3 shows the method to obtain the suitable value of WS. The size of the specimen is 100 mm ×100 mm × 2 mm, firstly. The distance from the specimen edge is defined as (1/2)WS which is the beginning position of the welding, as shown in Figure 3a. Figure 4 shows the macro profile of the solidification cracking under the different values of (1/2)WS during laser welding at welding speed of 1.0 m/min. Figure 5 shows the solidification cracking length under the different values of (1/2)WS during laser welding at welding speed of 1.0 m/min corresponding to those of Figure 4. When the value of (1/2)WS was less than 6 mm, the solidification cracking appeared and the length was more than 80 mm at both the front and backside of the specimen. While, there was no crack under the value of (1/2)WS more than 7 mm, as shown in Figures 4 and 5. According to the above principle, the value of WS could be set 12 mm in order to initiate crack at the beginning of the welding. … should read as… Figure 3 shows the method to obtain a suitable value of WS. The size of the specimen is 100 mm ×100 mm × 2 mm, initially. The distance from the specimen edge is defined as (1/2)WS, which is the beginning position of the welding, as shown in Figure 3a. Figure 4 shows the macro profile of the solidification cracking under the different values of (1/2)WS during laser welding at a welding speed of 1.0 m/min. Figure 5 shows the solidification cracking length under the different values of (1/2)WS during laser welding at a welding speed of 1.0 m/min corresponding to those of Figure 4. When the value of (1/2)WS was less than 6 mm, the solidification cracking appeared, and the length was more than 80 mm at both the front and backside of the specimen. While, there was no crack under the value of (1/2)WS more than 7 mm, as shown in Figures 4 and 5. According to the above principle, the value of WS could be set 12 mm to initiate crack at the beginning of the welding.
Response 25: The detailed revision is as follow:
In line 130-140, “Figure 3 shows the method to obtain the suitable value of WS. The size of the specimen is 100 mm ×100 mm × 2 mm, firstly. The distance from the specimen edge is defined as (1/2)WS which is the beginning position of the welding, as shown in Figure 3a. Figure 4 shows the macro profile of the solidification cracking under the different values of (1/2)WS during laser welding at welding speed of 1.0 m/min. Figure 5 shows the solidification cracking length under the different values of (1/2)WS during laser welding at welding speed of 1.0 m/min corresponding to those of Figure 4. When the value of (1/2)WS was less than 6 mm, the solidification cracking appeared and the length was more than 80 mm at both the front and backside of the specimen. While, there was no crack under the value of (1/2)WS more than 7 mm, as shown in Figures 4 and 5. According to the above principle, the value of WS could be set 12 mm in order to initiate crack at the beginning of the welding.” has been changed into “Figure 3 shows the method to obtain a suitable value of WS. The size of the specimen is 100 mm ×100 mm × 2 mm, initially. The distance from the specimen edge is defined as (1/2)WS, which is the starting position of the welding, as shown in Figure 3a. Figure 4 shows the macro profile of the solidification cracking under the different values of (1/2)WS during laser welding at a welding speed of 1.0 m/min. Figure 5 shows the solidification cracking length under the different values of (1/2)WS during laser welding at a welding speed of 1.0 m/min corresponding to those of Figure 4. When the value of (1/2)WS was less than 6 mm, the solidification cracking appeared, and the length was more than 80 mm at both the front and backside of the specimen. While, there was no crack under the value of (1/2)WS more than 7 mm, as shown in Figures 4 and 5. According to the above principle, the value of WS could be set 12 mm to initiate crack at the starting of the welding.
Point 26: Line 142: Figure 4. The macro profile of the solidification cracking of SUS310 stainless steel under the different values of (1/2)WS during laser welding at welding speed of 1.0 m/min… should read as… Figure 4. The macro profile of the solidification cracking of SUS310 stainless steel under the different values of (1/2)WS during laser welding at a welding speed of 1.0 m/min.
Response 26: The detailed revision is as follow:
In line 142, “Figure 4. The macro profile of the solidification cracking of SUS310 stainless steel under the different values of (1/2)WS during laser welding at welding speed of 1.0 m/min.” has been changed into “Figure 4. The macro profile of the solidification cracking of SUS310 stainless steel under the different values of (1/2)WS during laser welding at a welding speed of 1.0 m/min.”
Point 27: Line 145: Effect of (1/2)WS on the solidification cracking length during laser welding at welding speed of 1.0 m/min… should read as…: Effect of (1/2)WS on the solidification cracking length during laser welding at a welding speed of 1.0 m/min.
Response 27: The detailed revision is as follow:
In line 145, “Figure 5. Effect of (1/2)WS on the solidification cracking length during laser welding at welding speed of 1.0 m/min.” has been changed into “Figure 5. Effect of (1/2)WS on the solidification cracking length during laser welding at a welding speed of 1.0 m/min.
Point 28: Line 147: Then, the value of WF was assumed as 20 mm, 30 mm, 40 mm and 50 mm, respectively. Figure 6 shows the macroscopic appearance of the solidification cracking of SUS310 stainless steel under WS of 12 mm and the different values of WF during laser welding at welding speed of 1.0 m/min… should read as… Then, the value of WF was assumed as 20 mm, 30 mm, 40 mm, and 50 mm, respectively. Figure 6 shows the macroscopic appearance of the solidification cracking of SUS310 stainless steel under WS of 12 mm and the different values of WF during laser welding at a welding speed of 1.0 m/min.
Response 28: The detailed revision is as follow:
In line 147, “Then, the value of WF was assumed as 20 mm, 30 mm, 40 mm and 50 mm, respectively. Figure 6 shows the macroscopic appearance of the solidification cracking of SUS310 stainless steel under WS of 12 mm and the different values of WF during laser welding at welding speed of 1.0 m/min.” has been changed into “Then, the value of WF was assumed as 20 mm, 30 mm, 40 mm, and 50 mm, respectively. Figure 6 shows the macroscopic appearance of the solidification cracking of SUS310 stainless steel under WS of 12 mm and the different values of WF during laser welding at a welding speed of 1.0 m/min.”
Point 29: Line 154: However, under the certain condition, the solidification cracking susceptibility can be characterized by the cracking length… should read as… However, under a certain condition, the solidification cracking susceptibility can be characterized by the cracking length.
Response 29: The detailed revision is as follow:
In line 154, “However, under the certain condition, the solidification cracking susceptibility can be characterized by the cracking length.” has been changed into “However, under a certain condition, the solidification cracking susceptibility can be characterized by the cracking length.”
Point 30: Line 162: Figure 6. Macroscopic appearance of the solidification cracking of SUS310 stainless steel under WS of 12 mm and the different values of WF during laser welding at welding speed of 1.0 m/min… should read as… Figure 6. Macroscopic appearance of the solidification cracking of SUS310 stainless steel under WS of 12 mm and the different values of WF during laser welding at a welding speed of 1.0 m/min.
Response 30: The detailed revision is as follow:
In line 162, “Figure 6. Macroscopic appearance of the solidification cracking of SUS310 stainless steel under WS of 12 mm and the different values of WF during laser welding at welding speed of 1.0 m/min.” has been changed into “Figure 6. Macroscopic appearance of the solidification cracking of SUS310 stainless steel under WS of 12 mm and the different values of WF during laser welding at a welding speed of 1.0 m/min.”
Point 31: Line 165: Figure 7. Effect of WF on the solidification cracking length under WS of 12 mm during laser welding at welding speed of 1.0 m/min… should read as… Figure 7. Effect of WF on the solidification cracking length under WS of 12 mm during laser welding at a welding speed of 1.0 m/min. Response 31: The detailed revision is as follow:
In line 165, “Figure 7. Effect of WF on the solidification cracking length under WS of 12 mm during laser welding at welding speed of 1.0 m/min.” has been changed into “Figure 7. Effect of WF on the solidification cracking length under WS of 12 mm during laser welding at a welding speed of 1.0 m/min.
Point 32: Line 169: Figure 8 shows the effect of the chemical components (different stainless steels) on the solidification cracking rate during laser welding as welding speed of 1.0 m/min. The sequence of the solidification cracking rate under WS of 12 mm and WF of 40 mm is as follow: SUS310 > SUS316 > SUS304. Especially, the solidification cracking rate of SUS310 stainless steel was much higher than the others… should read as… Figure 8 shows the effect of the chemical components (different stainless steels) on the solidification cracking rate during laser welding at a welding speed of 1.0 m/min. The sequence of the solidification cracking rate under WS of 12 mm and WF of 40 mm is as follows: SUS310 > SUS316 > SUS304. Especially, the solidification cracking rate of SUS310 stainless steel was much higher than the others.
Response 32: The detailed revision is as follow:
In line 169, “Figure 8 shows the effect of the chemical components (different stainless steels) on the solidification cracking rate during laser welding as welding speed of 1.0 m/min. The sequence of the solidification cracking rate under WS of 12 mm and WF of 40 mm is as follow: SUS310 > SUS316 > SUS304. Especially, the solidification cracking rate of SUS310 stainless steel was much higher than the others.” has been changed into “Figure 9 shows the effect of the chemical components (different stainless steels) on the solidification cracking rate during laser welding at a welding speed of 1.0 m/min. The sequence of the solidification cracking rate under WS of 12 mm and WF of 40 mm is as follows: SUS310 > SUS316 > SUS304. Especially, the solidification cracking rate of SUS310 stainless steel was much higher than the others.
Point 33: Line 175: Figure 8. Effect of the chemical components on the solidification cracking rate during laser welding at welding speed of 1.0 m/min…should read as… Figure 8. Effect of the chemical components on the solidification cracking rate during laser welding at the welding speed of 1.0 m/min.
Response 33: The detailed revision is as follow:
In line 175, “Figure 8. Effect of the chemical components on the solidification cracking rate during laser welding at welding speed of 1.0 m/min.” has been changed into “Figure 9. Effect of the chemical components on the solidification cracking rate during laser welding at the welding speed of 1.0 m/min.”
Point 34: Line 180: The austenite structure and some lath ferrite structure could be observed from the microstructure images of SUS316 and SUS304 stainless steels welding, and further the amount of ferrite structure in SUS304 stainless steel seemed to be more than that of SUS316 stainless steel, as shown in Figure 9b, c, e and f… should read as…The austenite some lath ferrite structures could be observed from the microstructure images of SUS316 and SUS304 stainless steels welding; further, the amount of ferrite structure in SUS304 stainless steel seemed to be more than that of SUS316 stainless steel, as shown in Figures 9b, c, e, and f.
Response 34: The detailed revision is as follow:
In line 180, “The austenite structure and some lath ferrite structure could be observed from the microstructure images of SUS316 and SUS304 stainless steels welding, and further the amount of ferrite structure in SUS304 stainless steel seemed to be more than that of SUS316 stainless steel, as shown in Figure 9b, c, e and f.” has been changed into “The austenite some lath ferrite structures could be observed from the microstructure images of SUS316 and SUS304 stainless steels welding; further, the amount of ferrite structure in SUS304 stainless steel seemed to be more than that of SUS316 stainless steel, as shown in Figures 10b, c, e, and f.”
Point 35: Line 184: According to the welding metallurgy of stainless steel and previous researches, the microstructures of the weld bead are depended on the solidification mode[26], while the value of Creq/Nieq can determine the solidification mode of the welding… should read as…According to the welding metallurgy of stainless steel and previous researches, the microstructures of the weld bead are dependent on the solidification mode [26]. At the same time, the value of Creq/Nieq can determine the solidification mode of the welding.
Response 35: The detailed revision is as follow:
In line 184, “According to the welding metallurgy of stainless steel and previous researches, the microstructures of the weld bead are depended on the solidification mode[26], while the value of Creq/Nieq can determine the solidification mode of the welding.” has been changed into “According to the welding metallurgy of stainless steel and previous researches, the microstructures of the weld bead are dependent on the solidification mode[26]. At the same time, the value of Creq/Nieq can determine the solidification mode of the welding.”
Point 36: Line 190: Please change the word “A mode” with “A-mode.
Response 36: The detailed revision is as follow:
In line 190, “If the value of Creq/Nieq is lower than 1.25, the stainless steel solidifies in fully austenitic mode (A mode), that is, the microstructure is fully austenitic at the end of solidification.” has been changed into “If the value of Creq/Nieq is lower than 1.25, the stainless steel solidifies in fully austenitic mode (A-mode), that is, the microstructure is fully austenitic at the end of solidification.”
Point 37: Line 191: When the value of Creq/Nieq is between 1.25 and 1.48, the initial phase is austenite and then δ-ferrite is formed by reaction, so it is called AF mode… should read as… When the value of Creq/Nieq is between 1.25 and 1.48, the initial phase is austenite, and then δ-ferrite is formed by reaction, so it is called AF mode.
Response 37: The detailed revision is as follow:
In line 191 “When the value of Creq/Nieq is between 1.25 and 1.48, the initial phase is austenite and then δ-ferrite is formed by reaction, so it is called AF mode.” has been changed into “When the value of Creq/Nieq is between 1.25 and 1.48, the initial phase is austenite, and then δ-ferrite is formed by reaction, so it is called AF-mode.”
Point 38: Line 192: Since the ferrite forming elements, such as Cr, can segregate in the subgrain boundary so as to promote the formation of ferrite, the cracking susceptibility could be reduced to a certain extent… should read as… Since the ferrite forming elements, such as Cr, can segregate in the subgrain boundary to promote the formation of ferrite, the cracking susceptibility could be reduced to a certain extent.
Response 38: The detailed revision is as follow:
In line 192, “Since the ferrite forming elements, such as Cr, can segregate in the subgrain boundary so as to promote the formation of ferrite, the cracking susceptibility could be reduced to a certain extent.” has been changed into “Since the ferrite forming elements, such as Cr, can segregate in the subgrain boundary to promote the formation of ferrite, the cracking susceptibility could be reduced to a certain extent.”
Point 39: Line 196: Because the primary ferrite can solute higher impurities, like S and P, that is beneficial to restricting the partitioning of these elements to the interdendritic regions causing a good resistance to the solidification cracking. If the value of Creq/Nieq is greater than 1.95, it forms the fully ferrite during the solidification[27]… should read as… Because the primary ferrite can solute higher impurities, like S and P, that is beneficial to restricting the partitioning of these elements to the interdendritic regions, causing a good resistance to the solidification cracking. If the value of Creq/Nieq is greater than 1.95, it forms the full ferrite during the solidification[27].
Response 39: The detailed revision is as follow:
In line 196, “Because the primary ferrite can solute higher impurities, like S and P, that is beneficial to restricting the partitioning of these elements to the interdendritic regions causing a good resistance to the solidification cracking. If the value of Creq/Nieq is greater than 1.95, it forms the fully ferrite during the solidification[27].” has been changed into “Because the primary ferrite can solute higher impurities, like S and P, that is beneficial to restrict the partitioning of these elements to the interdendritic regions, causing a good resistance to the solidification cracking. If the value of Creq/Nieq is greater than 1.95, it forms the full ferrite during the solidification[27].”
Point 40: Line 200: Through calculation, the values of Creq/Nieq in SUS310, SUS316 and SUS304 stainless steels were 1.11, 1.53, and 1.91, respectively… should read as… Through calculation, the values of Creq/Nieq in SUS310, SUS316, and SUS304 stainless steels were 1.11, 1.53, and 1.91, respectively.
Response 40: The detailed revision is as follow:
In line 200, “Through calculation, the values of Creq/Nieq in SUS310, SUS316 and SUS304 stainless steels were 1.11, 1.53, and 1.91, respectively.” has been changed into “Through calculation, the values of Creq/Nieq in SUS310, SUS316, and SUS304 stainless steels were 1.11, 1.53, and 1.91, respectively.”
Point 41: Line 202: Obviously, the fully austenitic structure contributed to the more susceptibility compared with ferrite and austenitic mixed structure, as shown in Figure 9… should be corrected as… The fully austenitic structure contributed to the more susceptibility compared with ferrite and austenitic mixed structure, as shown in Figure 9.
Response 41: The detailed revision is as follow:
In line 202, “Obviously, the fully austenitic structure contributed to the more susceptibility compared with ferrite and austenitic mixed structure, as shown in Figure 9.” has been changed into “The fully austenitic structure contributed to the more susceptibility compared with ferrite and austenitic mixed structure, as shown in Figure 10.”
Point 42: Line 214: OM and SEM images of the microstructure of the cross-section of weld joint during laser welding at welding speed of 1.0 m/min… should read as…OM and SEM images of the microstructure of the cross-section of the weld joint during laser welding at a welding speed of 1.0 m/min.
Response 42: The detailed revision is as follow:
In line 214, “OM and SEM images of the microstructure of the cross-section of weld joint during laser welding at welding speed of 1.0 m/min.” has been changed into “OM and SEM images of the microstructure of the cross-section of the weld joint during laser welding at a welding speed of 1.0 m/min.”
Point 43: Line 218: Figure 10 shows the effect of welding speed on the solidification cracking rate of SUS310 and SUS304 stainless steels during laser welding using the Trapezoidal hot cracking test with WS of 12 mm. For SUS310 stainless steel, the solidification cracking rate was done under WF of 40 mm during laser welding at different welding speeds from 1.0 m/min to 2.0 m/min, since the cracking rate was already the relatively high during laser welding at welding speed of 1.0 m/min under WF of 20 mm. In addition, due to the relatively low solidification cracking rate in SUS304 stainless steel compared with that of SUS310 stainless steel, the value of WF was set 20 mm in order to measure and compare the cracking rate conveniently. The results showed that with the increase of laser welding speed, the solidification cracking rate of SUS310 stainless steel tended to decrease gradually, that is, the cracking susceptibility dropped slowly in the case of A mode solidification. In contrast, there was a tendency of increasing in the solidification cracking rate of SUS304 stainless steel with the increase of welding speed, that is, the cracking susceptibility augmented in the case of FA mode solidification…. Should be corrected as… Figure 10 shows the effect of welding speed on the solidification cracking rate of SUS310 and SUS304 stainless steels during laser welding using the Trapezoidal hot cracking test with WS of 12 mm. For SUS310 stainless steel, the solidification cracking rate was done under WF of 40 mm during laser welding at different welding speeds from 1.0 m/min to 2.0 m/min, since the cracking rate was already the relatively high during laser welding at a welding speed of 1.0 m/min under WF of 20 mm. Besides, due to the relatively low solidification cracking rate in SUS304 stainless steel compared with that of SUS310 stainless steel, the value of WF was set 20 mm to measure and compare the cracking rate conveniently. The results showed that with the increase of laser welding speed, the solidification cracking rate of SUS310 stainless steel tended to decrease gradually, that is, the cracking susceptibility dropped slowly in the case of A mode solidification. In contrast, there was a tendency of increasing in the solidification cracking rate of SUS304 stainless steel with the increase of welding speed, that is, the cracking susceptibility augmented in the case of FA mode solidification.
Response 43: The detailed revision is as follow:
In line 218, “Figure 10 shows the effect of welding speed on the solidification cracking rate of SUS310 and SUS304 stainless steels during laser welding using the Trapezoidal hot cracking test with WS of 12 mm. For SUS310 stainless steel, the solidification cracking rate was done under WF of 40 mm during laser welding at different welding speeds from 1.0 m/min to 2.0 m/min, since the cracking rate was already the relatively high during laser welding at welding speed of 1.0 m/min under WF of 20 mm. In addition, due to the relatively low solidification cracking rate in SUS304 stainless steel compared with that of SUS310 stainless steel, the value of WF was set 20 mm in order to measure and compare the cracking rate conveniently. The results showed that with the increase of laser welding speed, the solidification cracking rate of SUS310 stainless steel tended to decrease gradually, that is, the cracking susceptibility dropped slowly in the case of A mode solidification. In contrast, there was a tendency of increasing in the solidification cracking rate of SUS304 stainless steel with the increase of welding speed, that is, the cracking susceptibility augmented in the case of FA mode solidification.” has been changed into “Figure 12 shows the effect of welding speed on the solidification cracking rate of SUS310 and SUS304 stainless steels during laser welding using the ‘Trapezoidal hot’ cracking test with WS of 12 mm. For SUS310 stainless steel, the solidification cracking rate was done under WF of 40 mm during laser welding at different welding speeds from 1.0 m/min to 2.0 m/min, since the cracking rate was already the relatively high during laser welding at a welding speed of 1.0 m/min under WF of 20 mm. Besides, due to the relatively low solidification cracking rate in SUS304 stainless steel compared with that of SUS310 stainless steel, the value of WF was set 20 mm to measure and compare the cracking rate conveniently. The results showed that with the increase of laser welding speed, the solidification cracking rate of SUS310 stainless steel tended to decrease gradually, that is, the cracking susceptibility dropped slowly in the case of A-mode solidification. In contrast, there was a tendency of increasing in the solidification cracking rate of SUS304 stainless steel with the increase of welding speed, that is, the cracking susceptibility augmented in the case of FA-mode solidification.”
Point 44: Line 231: The grains in SUS310 stainless steel had a certain degree of refinement during the increase of welding speed, as shown in Figure 11a, b. While, for SUS304 stainless steel during laser welding, the amount of δ-ferrite in the microstructure was obviously less at welding speed of 2.0 m/min than those at 1.0 m/min, as shown in Figure 11c, d… should read as… The grains in SUS310 stainless steel had a certain degree of refinement during the increase of welding speed, as shown in Figure 11a, b. While, for SUS304 stainless steel during laser welding, the amount of δ-ferrite in the microstructure was less at a welding speed of 2.0 m/min than those at 1.0 m/min, as shown in Figure 11c, d.
Response 44: The detailed revision is as follow:
In line 231, “The grains in SUS310 stainless steel had a certain degree of refinement during the increase of welding speed, as shown in Figure 11a, b. While, for SUS304 stainless steel during laser welding, the amount of δ-ferrite in the microstructure was obviously less at welding speed of 2.0 m/min than those at 1.0 m/min, as shown in Figure 11c, d.” has been changed into “The grains in SUS310 stainless steel had a certain degree of refinement during the increase of welding speed, as shown in Figure 13a, b. While, for SUS304 stainless steel during laser welding, the amount of δ-ferrite in the microstructure was less at a welding speed of 2.0 m/min than those at 1.0 m/min, as shown in Figure 13c, d.”
Point 45: Line 235: With the increase of welding speed, the welding cooling rate increases, causing grain refinement. During solidification, the grain refinement can not only promote the refill and healing of liquid phase more effectively, but also increase the number of grain boundaries, further preventing harmful low melting-point compound from segregating on grain boundaries [29]… should be… With the increase of welding speed, the welding cooling rate increases, causing grain refinement. During solidification, the grain refinement can not only promote the refill and healing of liquid phase more effectively but also increase the number of grain boundaries, further preventing harmful low melting-point compound from segregating on grain boundaries.
Response 45: The detailed revision is as follow:
In line 235, “With the increase of welding speed, the welding cooling rate increases, causing grain refinement. During solidification, the grain refinement can not only promote the refill and healing of liquid phase more effectively, but also increase the number of grain boundaries, further preventing harmful low melting-point compound from segregating on grain boundaries [29].” has been changed into “Theoretically, when the power is fixed during laser welding, the welding heat input calculated using the power divided by the welding speed will decrease with the increase of welding speed, resulting in the appearance of a relatively higher cooling rate, and further causing grain refinement[29]. While, the heat inputs are 108 J/mm for a welding speed of 1.0 m/min, 92 J/mm for a welding speed of 1.5 m/min and 84 J/mm for a welding speed of 2.0 m/min by calculation, respectively. Thus, the cooling rate tended to increase with increasing welding speed from 1.0 m/min to 2.0 m/min in this experiment during laser welding, inducing to the phenomenon that the grain size slowed down. During solidification, the grain refinement can not only promote the refill and healing of liquid phase more effectively but also increase the number of grain boundaries, further preventing harmful low melting-point compound from segregating on grain boundaries[30].
Point 46: Line 248: Figure 10. Effect of the welding speed on the solidification cracking rate during laser welding… should read as… Effect of the welding speed on the solidification cracking rate during laser welding.
Response 46: The detailed revision is as follow:
In line 248, “Figure 10. Effect of the welding speed on the solidification cracking rate during laser welding.” has been changed into “Figure 12. Effect of the welding speed on the solidification cracking rate during laser welding.
Point 47: Line 251: OM image of the microstructure of the cross-section of weld joint… should read as… OM image of the microstructure of the cross-section of the weld joint.
Response 47: The detailed revision is as follow:
In line 251, “Figure 11. OM image of the microstructure of the cross-section of weld joint. (a) OM image in SUS310 at welding speed of 1.0 m/min; (b) OM image in SUS310 at welding speed of 2.0 m/min; (c) OM image in SUS304 at welding speed of 1.0 m/min; (d) OM image in SUS304 at welding speed of 2.0 m/min.” has been changed into “Figure 13. OM image of the microstructure of the cross-section of the weld joint. (a) OM image in SUS310 at the welding speed of 1.0 m/min; (b) OM image in SUS310 at the welding speed of 2.0 m/min; (c) OM image in SUS304 at the welding speed of 1.0 m/min; (d) OM image in SUS304 at the welding speed of 2.0 m/min.”
Point 48: Line 258: The new test method called the Trapezoidal hot cracking test was developed to evaluate the solidification cracking susceptibility using the solidification cracking rate which was the ratio of the solidification cracking length to the weld bead length under the suitable values of WS and WF… should be…The new test method called the Trapezoidal hot cracking test was developed to evaluate the solidification cracking susceptibility using the solidification cracking rate which was the ratio of the solidification cracking length to the weld bead length under the suitable values of WS and WF.
Response 48: The detailed revision is as follow:
In line 258, “The new test method called the Trapezoidal hot cracking test was developed to evaluate the solidification cracking susceptibility using the solidification cracking rate which was the ratio of the solidification cracking length to the weld bead length under the suitable values of WS and WF.” has been changed into “The new test method called the “Trapezoidal hot” cracking test was developed to evaluate the solidification cracking susceptibility using the solidification cracking rate which was the ratio of the solidification cracking length to the weld bead length under the suitable values of WS and WF.”
Point 49: Line 261: The sequence of the solidification cracking susceptibility was as follow: SUS310 > SUS316 > SUS304 during laser welding at welding speed of 1.0 m/min… should read as… The sequence of the solidification cracking susceptibility was as follow: SUS310 > SUS316 > SUS304 during laser welding at a welding speed of 1.0 m/min.
Response 49: The detailed revision is as follow:
In line 261, “The sequence of the solidification cracking susceptibility was as follow: SUS310 > SUS316 > SUS304 during laser welding at welding speed of 1.0 m/min, because the fully austenitic structure appeared in the weld joint of SUS310 stainless steel, and the ferrite and austenitic mixed structures produced in SUS316 and SUS304 stainless steel, moreover, the latter had relatively more ferrite.” has been changed into “The sequence of the solidification cracking susceptibility was as follow: SUS310 > SUS316 > SUS304 during laser welding at a welding speed of 1.0 m/min, because the fully austenitic structure appeared in the weld joint of SUS310 stainless steel, and the ferrite and austenitic mixed structures produced in SUS316 and SUS304 stainless steel, moreover, the latter had relatively more ferrite.”
Point 50: Line 267: This is due to the grain refinement for the former and a decrease in the amount of ferrite for the latter…. should be… It is due to the grain refinement for the former and a decrease in the amount of ferrite for the latter.
Response 50: The detailed revision is as follow:
In line 267, “This is due to the grain refinement for the former and a decrease in the amount of ferrite for the latter.” has been changed into “It is due to the grain refinement for the former and a decrease in the amount of ferrite for the latter.”
